# Ectopic expression of sericin enables efficient production of ancient silk with structural changes in silkworm

Xuedong Chen[1,2,4], Yongfeng Wang [1,2,4], Yujun Wang[3,4], Qiuying Li[1,2], Xinyin Liang[1,2], Guang Wang[1,2], Jianglan Li[1,2], Ruji Peng[1,2], Yanghu Sima[1,2] & Shiqing Xu [1,2] ✉

*Bombyx mori* silk is a super-long natural protein fiber with a unique structure and excellent performance. Innovative silk structures with high performance are in great demand, thus resulting in an industrial bottleneck. Herein, the outer layer sericin SER3 is ectopically expressed in the posterior silk gland (PSG) in silkworms via a *piggyBac*-mediated transgenic approach, then secreted into the inner fibroin layer, thus generating a fiber with sericin microsomes dispersed in fibroin fibrils. The water-soluble SER3 protein secreted by PSG causes P25's detachment from the fibroin unit of the Fib-H/Fib-L/P25 polymer, and accumulation between the fibroin layer and the sericin layer. Consequently, the water solubility and stability of the fibroin-colloid in the silk glandular cavity, and the crystallinity increase, and the mechanical properties of cocoon fibers, moisture absorption and moisture liberation of the silk also improve. Meanwhile, the mutant overcomes the problems of low survival and abnormal silk gland development, thus enabling higher production efficiency of cocoon silk. In summary, we describe a silk gland transgenic target protein selection strategy to alter the silk fiber structure and to innovate its properties. This work provides an efficient and green method to produce silk fibers with new functions.

The beautiful silk fibers produced by the silkworm (*Bombyx mori*) have excellent performance and are an easily available renewable protein material. The toughness of silk fiber and its unusual combination of high strength and expansibility have not been surpassed by synthetic materials to date[1,2]. The silk gland (SG) in silkworm larvae is the most efficient insect organ for protein synthesis and exocrine secretion; it can synthesize 20–35% of its own weight in protein in approximately 1 week in 5th instar larvae[3]. The concentration of aqueous silk protein solution in the SG cavity is as high as 30%. This fiber processing unit, which maintains a metastable state of ultra-high-level protein, is difficult to

recapitulate through modern textile engineering technology[4,5]. Simulating the biological template of SG has emerged as a new research direction for developing high-performance, multifunctional protein fiber materials through green chemical processing. The multifunctional materials processed from silk, such as hydrogels, fibers, sponges, films, and other forms, has been used in many applications, such as medical materials, electronic information, and fine chemicals, thus demonstrating broad application potential[1,2,6].

Although many important insights in the synthesis and self-assembly of silk proteins have been obtained in the past 10 years[7–13],

[1]School of Biology and Basic Medical Sciences, Suzhou Medical College, Soochow University, Suzhou 215123, China. [2]National Engineering Laboratory for Modern Silk, Soochow University, Suzhou 215123, China. [3]Guangxi Key Laboratory of Beibu Gulf Marine Biodiversity Conservation, College of Marine Sciences, Beibu Gulf University, Qinzhou 535011, China. [4]These authors contributed equally: Xuedong Chen, Yongfeng Wang, Yujun Wang. ✉e-mail: szsqxu@suda.edu.cn

understanding remains lacking regarding the mechanism of the metastability of ultra-high concentration aqueous solutions of Fib-H/Fib-L/P25 polymers in SGs. Many advances and engineering applications have extended the functions of silk fibers, including silk processing by chemical or physical methods, and obtaining biomaterials for many purposes by modulating the self-assembly properties of silk fibroin[14–19]. However, these achievements have been based on the reprocessing and transformation of the natural silk structure. In the future, major advancements in related technologies and achievements will depend on breakthroughs in altering the natural silk structure.

The germplasm resources for mutant genes associated with silkworm cocoon silk purification have a long history of thousands of years, and the cultivation of hybrid varieties has also been performed for hundreds of years. These efforts have greatly contributed to optimizing the fiber characteristics of silk. However, owing to a bottleneck in the homogenization of silkworm varieties, the silk fibers produced by thousands of silkworm varieties worldwide have nearly the same composition, structure and characteristics[20,21]. An attractive method is to directly integrating special functional fiber protein genes into the silkworm genome has been described to aid in achieving high-efficiency expression in SGs, such as the expression of a high-strength spider silk protein gene in silkworm SGs to obtain silk fibers with improved mechanical properties[22–25], and the expression of fusions of optical functional protein and silk protein to obtain photoelectric silk or fluorescent silk[26,27]. Although the efforts to express and secrete exogenous proteins in the SGs of silkworms through transgenic technology to date have yielded many successful examples of genetic alterations. major challenges remain in substantially improving the expression efficiency of foreign proteins while maintaining the cocoon silk yield, particularly the expression of high molecular weight proteins (~100 kDa) in the posterior silk glands[23,24,28–32].

In this work, we implemented a new transgenic strategy to express our own water-soluble non-fibrin sericin in the posterior silk gland of silkworms, addressing the bottleneck due to silk gland deformity and low silk production rates in transgenic silkworms. Interestingly, the outer layer sericin SER3 of silk is secreted into the inner fibroin layer through the transgenic method, and a new structural fiber with non-fibrous sericin microsomes dispersed in fibroin fibrils is obtained. Importantly, we find that the P25 protein detached from the fibroin unit of Fib-H/Fib-L/P25 polymer, and accumulated on the surface of fibroin, and the mechanical properties of cocoon fibers, moisture absorption and moisture liberation of the silk were also improved. Thus, we propose that this work provides new ideas for silk protein fiber molecular design.

## Results

### A genetically modified silkworm mutation system to alter silk fiber structure

The structure and formation of silkworm silk fiber is shown in Fig. 1a. The water-soluble Sericin III (SER3) protein secreted by the anterior part of the middle silk gland (MSG) and wrapped in the outermost layer of the silk is not present in the fibroin fibrils. The technical strategy of this study involved specifically expressing SER3 recombinant protein in silkworm PSG cells to achieve secretion into the PSG lumen and the incorporation of water-soluble SER3 into the silk fibroin colloidal solution, thus altering the metastable state of silk fibroin protein polymers and further affecting the structure of silk fibers after self-assembly (Fig. 1b, c). Using red fluorescence in the eyes and green fluorescence in silk fibers as markers, after six consecutive generations of screening, we obtained the SER (*Ser3'/Ser3'*) mutant system (Supplementary Fig. 1b–d). In the PSG cells of the SER larvae, the mRNA and SER3

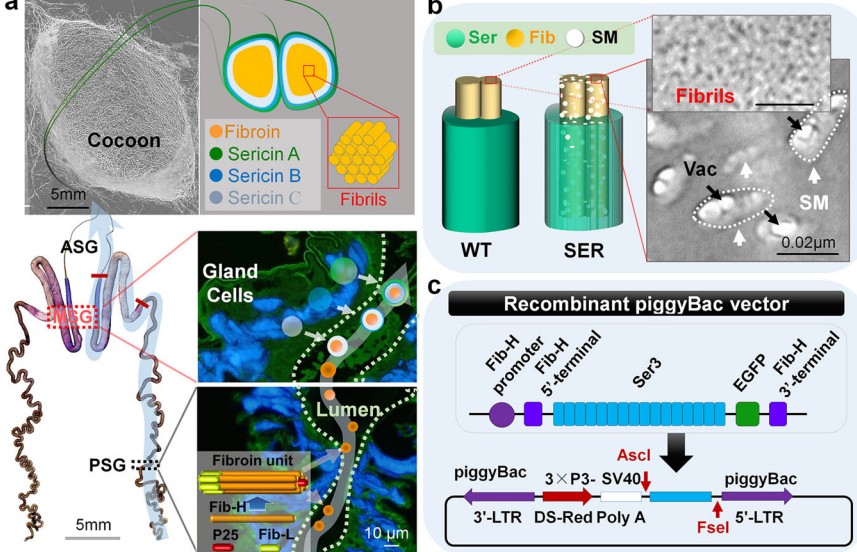

**Fig. 1 | Construction of transgenic silkworms. a** Schematic diagram of SGs producing cocoon silk. The silk fibrils are long chains formed by Fib-H/Fib-L/P25 polymer fibroin units synthesized by the PSG. The silk fibroin heavy chain protein (Fib-H), silk fibroin light chain protein (Fib-L), and P25 protein synthesized by PSG cells are secreted into the lumen as fibroin units and then are transferred to the MSG as a metastable high-concentration aqueous colloid. In the MSG lumen, the aqueous colloid of silk fibroin is surrounded by sericin C (sericin 1 is the main composition), which is secreted by the back end and middle part of the MSG, and then is surrounded by sericin B (mixed sericin) and sericin A (sericin 3 is the main composition), which are secreted by the middle and anterior part of the MSG. **b** Technical strategy. Efficient transgenic expression of SER3 recombinant protein in silkworm PSG cells, to achieve secretion into the PSG lumen and to incorporate water-soluble sericin 3 recombinant protein into the silk fibroin colloidal solution, to

alter the metastable state of silk fibroin protein polymers and further affect the silk fiber structure after self-assembly. SM, sericin microsomes expressed in the PSGs and incorporated into the cocoon silk fibrils. Vac, vacuoles in SM. SER, transgenic mutant system for expression of the sericin 3 recombinant gene in the PSG. WT, wild type. **c** Transgenic piggyBac vector. To enhance the expression and secretion of sericin 3protein (SER3) by PSG cells, the *Fib-H* gene promoter sequence and 1416 bp of its base sequence containing the signal peptide were introduced upstream of the sericin 3 gene (*Ser3*) sequence with a length of 3120 bp (Supplementary Sequence 1). The *EGFP* reporter gene sequence and the 333 bp base sequence at the 3' end of the *Fib-H* gene were connected downstream of the *Ser3* gene sequence. Moreover, an artificial promoter, 3 × P3, composed of three tandem *PAX-6* transcription factor binding sequences, was specifically expressed in the silkworm eyes and nervous system, and was used to regulate the *RFP* reporter gene.

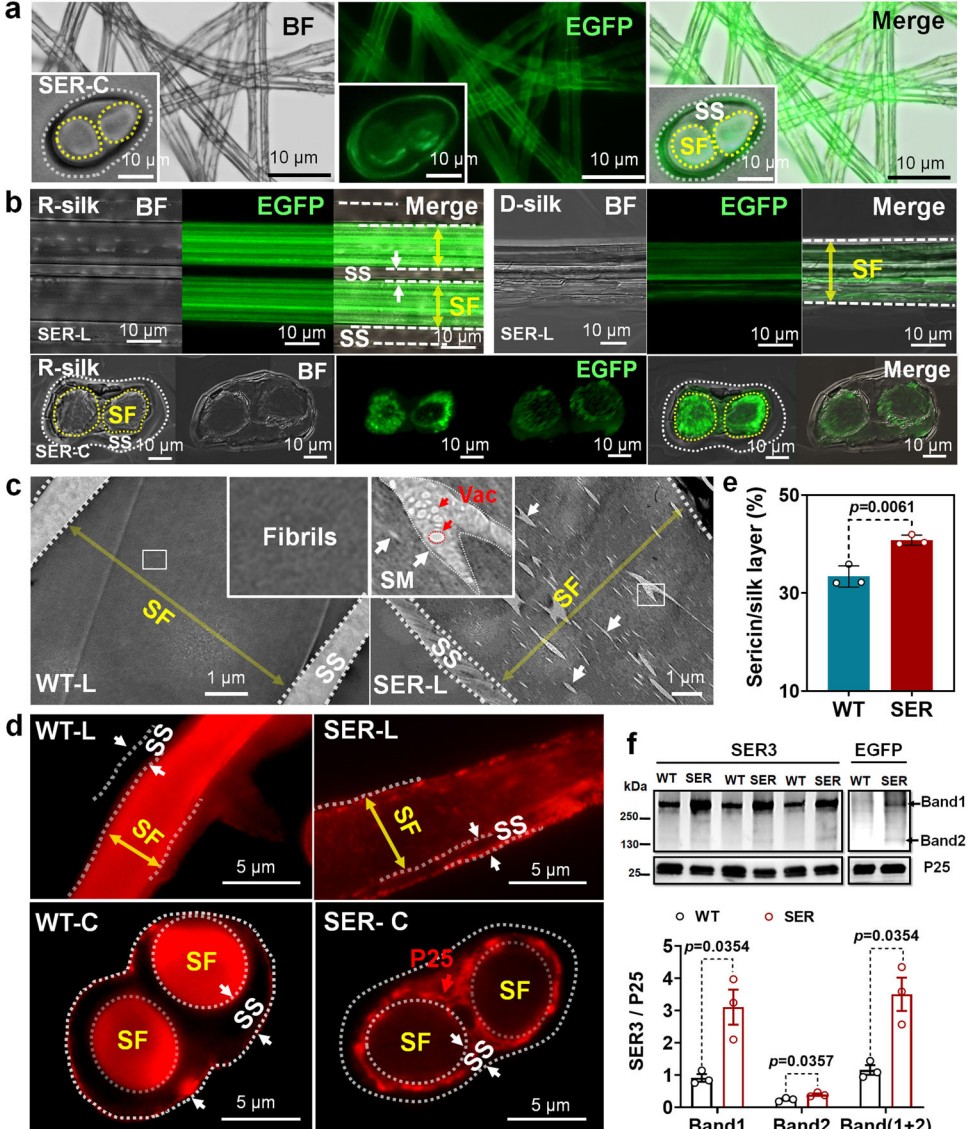

**Fig. 2 | Distribution of SER3 recombinant protein synthesized by the PSG in cocoon silk and its effect on fiber structure.** Images of silk fibers observed under **a** fluorescence microscope and **b** laser confocal microscope. EGFP fluorescence localization of SER3 recombinant protein synthesized by the PSG in silk fiber. **c** Transmission electron micrograph of a longitudinal silk fiber section. SER3 protein synthesized by the PSG is dispersed in the fibrils of the silk fiber. **d** Immunofluorescence localization of P25 protein in silk fiber. in Fig. 2a to Fig. 2d: WT-L and SER-L, longitudinal section of silk fiber; WT-C and SER-C, cross-section of silk fiber; R-silk, raw silk; D-silk, degummed silk. SF, fibroin layer of silk. SS, sericin

layer of silk. SM, sericin protein microsomes in the silk fibroin fibrils. Vac, vacuoles. **e** Sericin content of cocoon silk was determined by a classical degumming method. Data were presented as mean ± SEM, and the unpaired *t*-test analysis was used for the comparison between the two groups. *n* = 3 cocoons. **f** The content of SER3 protein in cocoon silk is determined by western blotting; P25 is an internal reference. *n* = 3. EGFP localization showed that the recombinant SER3 protein had haploid and dimer 2 types, and dimer was the main type. Data were presented as mean ± SEM. Image data are representative of three independent experiments unless otherwise stated.

protein expressed by the *Ser3* gene were detected, and the results were consistent with those from the screening of the *RFP/EGFP* reporter genes (Supplementary Fig. 1e, f). Tail-PCR detection revealed that a single copy of the *piggyBac* transposon was inserted into the *Bombyx mori* genome at the non-functional gene sequence at Chr.23 (scaf12: 4699379...4699384) (Supplementary Fig. 1g). Many sericin microsomes (SM) were present in the fibroin of silk fibers produced by SER silkworms, and vacuoles were observed in the SM (Fig. 1b). Continued investigation for 12 generations revealed that the mutant silkworms showed stable growth and development, and the production efficiency of cocoon silk was significantly higher than that of the wild type (WT). We observed no SG shortening, deformities, or decreased individual survival rates, which are common problems in SG transgenic silkworms

(Supplementary Fig. 2). From the perspective of sericulture, our findings demonstrate that the transgenic silkworm SGs have superior production performance.

## Mutant silkworms show silk fiber structure rearrangement and performance improvement

The silk fibers produced by the mutant silkworms exhibited green fluorescence from an EGFP fusion with SER3. As observed from a cross-section of the silk fiber, the fluorescence distribution was uneven, and strong fluorescence appeared between the fibroin and sericin layers (Fig. 2a). The results of laser confocal microscopy more clearly indicated that the recombinant SER3 protein in the fibroin area was present in particles of different sizes. The measurable particle size was 0.05–0.50 μm, and the particles were unevenly distributed, and

primarily found between the fibroin layer and sericin layer, then among the fibrils of silk fibroin (Fig. 2b). The longitudinal section of the silk fiber was observed by transmission electron microscopy (TEM). Large amounts of SER3-EGFP microsomes (SM) in the SER group were dispersed in the silk fibroin area. The SM shape was rain-thread-like or fusiform, and the shape was altered in the same direction as the movement of silk protein colloid under squeezing during the spinning process in mature larvae. Notably, in SM, vacuoles of low-density silk protein aqueous solutions of different sizes and shapes were observed (Fig. 2c). The cross-sectional TEM images further confirmed the presence of SM and vacuoles in the mutant silk fibers (Supplementary Fig. 3). After use of the classical cocoon silk degumming method, the sericin content in cocoon silk was determined. The percentage of sericin in cocoon silk in the SER group was 7.39% higher than that in the WT group (Fig. 2e), indicating an increase in 21.8%. Western blotting was used to determine the SER3 content in cocoon silk with P25 as an internal reference. The SER3 content (native plus recombinant SER3) in the mutant was 3 times that in the WT (Fig. 2f). Our results indicated that the PSG of the mutant silkworm synthesized the SER3 protein very efficiently and successfully secreted it into the silk fiber.

Using immunofluorescence, we observed that the P25 protein in the WT silk fiber was evenly distributed in the silk fibril area, whereas in the silk fibers of SER, almost all P25 had transferred to the outside of the silk fibrils and was unevenly distributed between the silk fibroin layer and the sericin layer, with different microsome sizes (Fig. 2d). Our findings suggested that P25 in SER silk fibers was separate from the silk protein comprising Fib-H/Fib-L/P25 polymers, thus indicating that the ordered fibril structure in the silk protein was greatly altered by the influence of the SER3 protein synthesized in the PSG.

The amino acid composition of silk fiber was analyzed. We observed no difference in amino acid composition between the cocoon silk of SER and WT. However, the silk fiber containing sericin protein showed changes in the relative content of a variety of amino acids, such as increased relative content of serine and aspartic acid and decreased relative content of glycine, alanine, and tyrosine. In the silk fiber (fibroin) for textile raw materials after removal of the outer sericin, the content of alanine increased by only 1.7% (29.9% in WT versus 30.41% in SER), and the relative content of other amino acids scarcely changed (Supplementary Table 2), because the amino acid composition of Fib-H/Fib-L/P25 polymers of silk fibroin is the same as that of SER3, and the relative content is also similar (Supplementary Table 3).

The mechanical properties of fibroin fibers after the removal of the outer sericin were analyzed. The stress and strain curve indicated that the tensile initial modulus of the SER group increased significantly, from 73.48 MPa to 110.93 MPa, a value 1.51 times higher than that in the WT group (Fig. 3a). The maximum stress level (Fig. 3b) and Young's modulus (Fig. 3d) in the SER group were also significantly higher than those in the WT group. Only the maximum elastic modulus had no statistically significant change (Fig. 3c). The moisture absorption and liberation performance of fibroin fibers showed significant improvements in the SER group. The moisture absorption curve (Fig. 3e) and moisture liberation curve (Fig. 3g) of silk fiber in the SER group were highly similar to those in the WT group. The rates of regaining moisture absorption and moisture liberation were 22.0% and 8.0% higher, respectively, than those in the WT group. The moisture absorption rate and moisture liberation rate within 0–1 min was 142.5–139.4% (Fig. 3f) and 165.5–164.1% (Fig. 3h) of those of the WT group, respectively.

SEM characterization indicated that the adhesion between silk fibers in the SER cocoon silk layer was closer, and the pores were smaller than those in the WT (Fig. 3i). After the removal of sericin with the alkali method, the surfaces of fibroin fibers in the SER group were smoother, and less fibril damage was observed than that in the WT (Fig. 3j). The results showed that the silk fibers in the SER group were less damaged by degumming than those in the WT group.

According to the crystal peak position of cocoon silk, the crystal diffraction peaks of silk fibroin fibers were detected at approximately 9.0°, 20.4°, and 29.1° (Fig. 3k). The calculated relative crystallinity results showed that the crystallinity of fibroin fibers in WT and SER groups was 36.62% and 42.29% respectively (Fig. 3l), thus indicating greater crystallinity of fibroin fibers in the SER group.

SAXS test results revealed two-dimensional images close to a double wedge shape (Fig. 3m), in which the short diameter in the SER group was longer than that in the WT group, thus indicating that both SER and WT cocoon silk fibers are anisotropic, but the electron density changes before and after X-ray transmission of the two materials differed. The scattering intensity curve showed a significant difference in discrete intensity in the angle range of angle 0.1° −0.6° (Fig. 3n), thus indicating that the mutant and WT cocoon silk differed in electron density in the crystalline and amorphous regions of the periodic structure (e.g., fibroin fibrils) at the nanoscale.

## Enhanced water solubility and stability of the silk fibroin colloid in the mutant PSG

Frozen sections were used to observe the SGs with the most vigorous stage of silk protein synthesis in the 5th instar 3rd day larvae, and the distribution of SER3 was assessed via the EGFP fusion protein (Fig. 4a). In the PSG lumen of the mutant larvae, we observed fluorescent particles of different sizes and shapes, with diameters of several micrometers (1–5 μm), scattered in a liquid comprising a grid of bubbles. The water-soluble EGFP-SER3 fusion protein distributed in the silk protein aqueous solution also entered the fibroin mass in an aggregated state. In the MSG lumen, the green fluorescence was distributed in both the fibroin and the sericin, but the fluorescence was stronger in the boundary area between the silk fibroin layer and the sericin layer. Notably, the fluorescent particles increased to tens of micrometers (10–50 μm) in diameter, and the bubble grid-like characteristics of the liquid distribution in the PSG lumen disappeared. The fluorescence distribution pattern in the ASG lumen indicated that the fluorescence in the outer layer of sericin was weak, and the distribution of fluorescent particles in the inner layer of silk fibroin tended to be uniform, but the sizes remained different, and the diameter decreased to 1–5 μm. On the microvilli in the MSG lumen and ASG, droplet-like green fluorescence was observed with a higher intensity than that in the sericin of the middle layer. With the movement of silk protein from the PSG to the ASG via the MSG, the aqueous solution of EGFP-SER3 fusion protein was incorporated into the forming fibroin mass, and the colloidal aggregation state of sericin (SER3) significantly changed, appearing in size and shape different fluid sericin microsomes. The structure and morphology of the fluid SER3 protein microsomes are shown in Fig. 2b and Supplementary Fig. 3.

TEM was used to observe the substructure of the SG cells and the secretion of silk protein in the 5th instar larvae (Fig. 4b). The organelles of the mutant PSG cells were normal, and appeared to be identical to those in the WT, with the abundant rough endoplasmic reticulum, Golgi apparatus, mitochondria, and other subcellular structures, thus indicating normal protein synthesis. The significant difference was that in the mutant PSG cells, the storage silk protein layer was thinner than that in the WT cells, and the amount of fibroin secreted into the glandular cavity was much greater. Few spherical aggregates of fibroin mass were observed in the lumen, and the silk protein colloids were more evenly distributed. Thus, the SER3 protein expression in the PSG improved the water solubility of the silk fibroin colloid.

The gene transcription levels of *EGFP*, *Ser3*, and the silk fibroin components *Fib-H*, *Fib-L*, and *P25* in different parts of the SG cells were measured (Fig. 4c–e). The PSG cells of the 5th instar larvae of the mutant efficiently expressed the *Ser3* gene, which is specifically

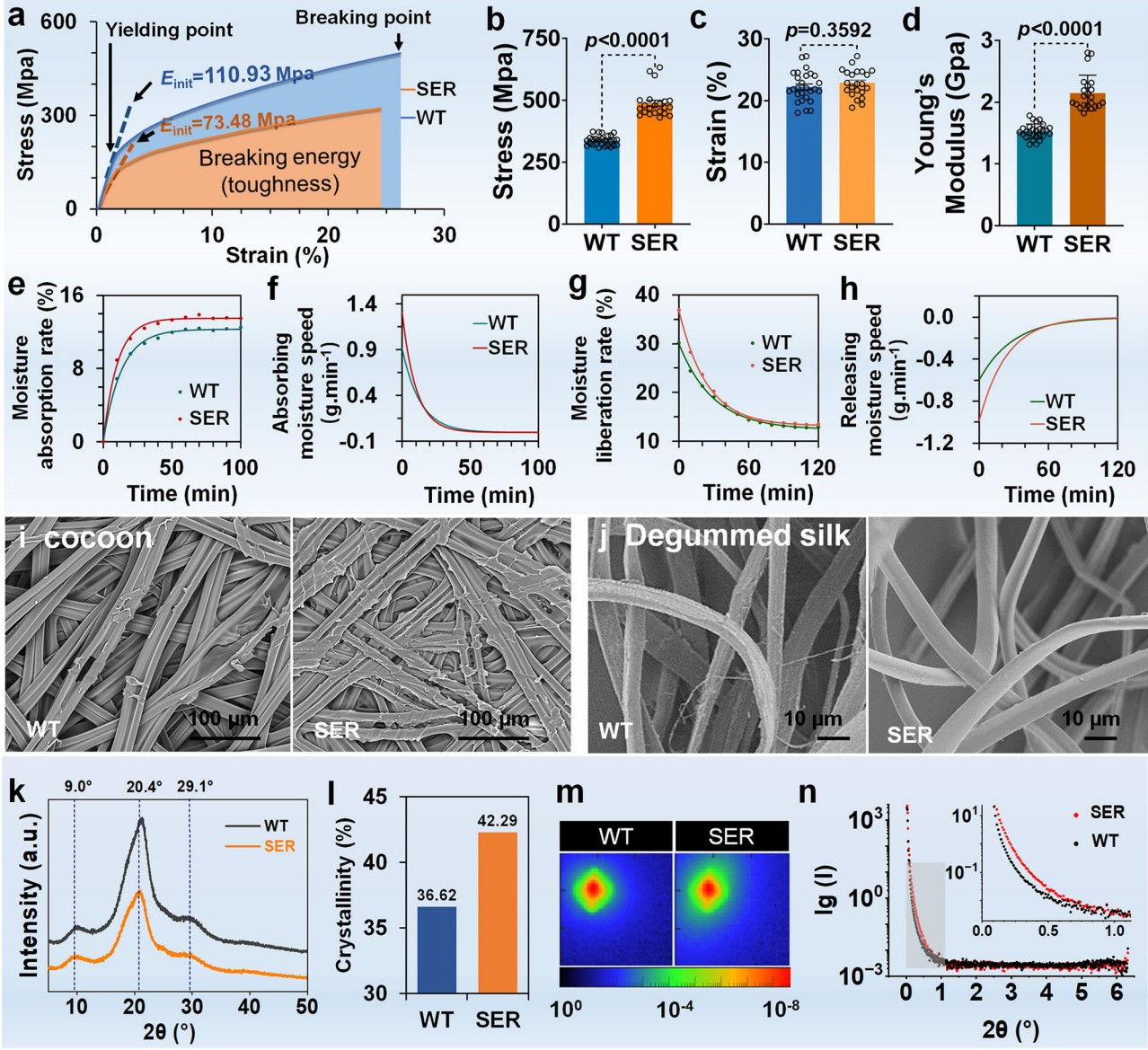

**Fig. 3 | The SER3 protein secreted by the PSG improves the physical properties of silk fibers. a–d** Fiber mechanical properties. Reeling 20/22 dtex raw silk from cocoons (20 cocoons of WT or SER: one sample was taken every 3–4 meters between 100–200 m to determine the mechanical properties. $n = 22$ cocoons in SER and n = 27 cocoons in WT. **a** Stress and strain curve. **b** Stress level. **c** Modulus of elasticity. **d** Young's modulus. Data were presented as mean ± SEM, and the unpaired t test analysis was used in (**b-d**). **e–h** Moisture absorption and desorption performance. The monofilament extracted from cocoons was boiled with 0.2% sodium carbonate for 30 min to remove the outer sericin protein and obtain textile fibroin fiber. **e** Moisture absorption rate (constant temperature and humidity conditions: 20 °C ± 2 °C, R.H. 65% ± 3%). **f** Moisture absorption speed. The fitted

curve equations of WT and SER fibroin fiber samples are v = 12.276−12.163e-0.0756t, R2 = 0.9966; v = 13.470−13.370e-0.0980t, R2 = 0.9954. t, time. **g** Moisture liberation rate (constant temperature and humidity conditions: 20 °C ± 2 °C, R.H. 100%). **h** Moisture release speed. The fitted curve equations of WT and SER fibroin fiber samples are v = 12.353 + 17.619e- 0.03381t, R2 = 0.9977; v = 13.184 + 23.552e-0.04187t, R2 = 0.9990. t, time. **i, j** Scanning electron microscopy (SEM) characterization of cocoon (**i**) and fibroin fiber (**j**). Image data are representative of three independent experiments unless otherwise stated. **k** XRD pattern of cocoon silk. **l** Crystallinity of cocoon silk. **m** SAXS diffractogram of cocoon silk. **n** SAXS diffraction data. Data were presented as mean ± SEM. $n = 3$ samples in (**e–n**).

expressed in the WT silkworm MSG (the anterior and middle parts of the MSG, MA, and MM). In the posterior parts of the PSG (PP) cells, the transcription level of the *Ser3* gene reached that in MM cells. Notably, the mRNA of the *Ser3* gene was detected in the posterior parts of the MSG (MP) cells of SER 5th instar larvae, although the transcription level was only 1–5% that of PSG cells (Fig. 4c–e), similarly to the fibroin genes expressed in the MP cells of both WT and SER 5th instars (Fig. 4c). Our results indicated that the *Fib-H* promoter used by the transgenic mutant expressed the *Ser3* and *EGFP* genes in MP cells and accounted for the strong green fluorescence observed in the outer sericin in the MSG lumen in Fig. 4a.

## Discussion

### The PSG expresses water-soluble sericin enabling the sustainable production of silk fibers with changes in their structure and properties

SER3 is wrapped in the outer layer of silkworm cocoon silk fibers, does not exist in the fibroin layer in its natural state, and does not contact the silk fibrils[3]. SER3 is a water-soluble protein[1] and is completely dissolved by hot water during the silk reeling process. The amino acid composition of the SER3 protein was the same as that of silk fibroin, and the relative amino acid content was also highly similar, thus avoiding imbalances in amino acid supply in the mutant PSG cells. The

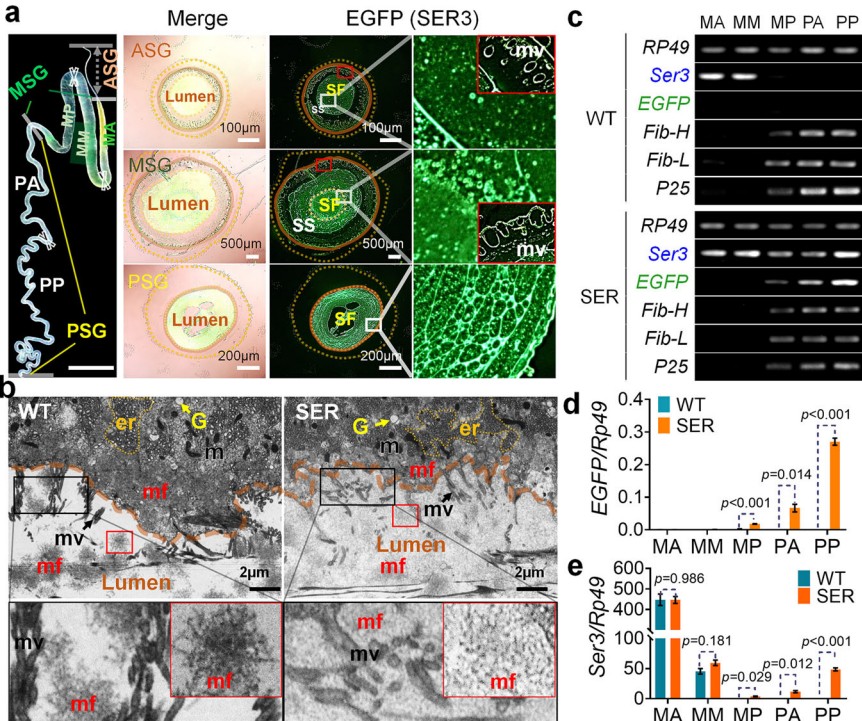

**Fig. 4 | Synthesis and secretion of silk proteins in SGs of mutant 5th instar larvae. a** Frozen section of the SG cross-section. The state of the SER3 protein secreted into the lumen of the SG, was detected by EGFP-fusion expression. **b** Transmission electron micrograph of the PSG. Fig. 4a and b: SF, silk fibroin layer; SS, silk sericin layer; er, endoplasmic reticulum; G, Golgi apparatus; m, mitochondrion; mf, fibroin mass; mv, microvilli. **c** Semi-quantitative PCR and **d, e** qRT-PCR to detect the mRNA levels of EGFP, SER3, and silk fibroin Fib-H, Fib-L, and P25 genes in cells in different parts of the SG. MA, MM, and MP show the anterior, middle, and posterior parts of the MSG, respectively. PA and PP show the anterior and posterior parts of the PSG, respectively. For **d** and **e**, Holm−Sidak t-test analysis was used and the p value obtained was the adjusted p value. Data were presented as mean ± SEM. n = 3 samples. Each tissue sample was collected from three female individuals, and each sample was measured three times. Image data are representative of three independent experiments unless otherwise stated.

ratio of the number of cysteine molecules in the amino acid residues of the SER3 protein (0.50 %) was intermediate between that of Fib-H (0.10 %) and Fib-L (1.10 %) (Supplementary Table 2), and may possibly form disulfide bonds and combine with Fib-H and other silk fibroin in the PSG. Meanwhile, P25 was able to combine with the NTD of recombinant SER3 protein, as with Fib-H. Herein, the recombinant *Ser3* gene specifically expressed by the silkworm MSG was expressed in the PSG, and the recombinant SER3 protein appeared in the long cocoon silk fibers composed of silk fibrils. The recombinant SER3 mutant was unevenly distributed in the fibroin (Fig. 4a), thus indicating that the recombinant SER3 could not form disulfide bonds and combine with Fib-H and Fib-L in the PSG. In the silk fibers produced by the mutant (Fig. 2), we observed SER3 protein microsomes dispersed among the silk fibrils in many different droplet sizes, thus indicating that the fibrils were not broken but had a partly changed arrangement. In the PSG of the SER silkworm larvae, less fibroin mass was retained in the gland cells, and we observed few spheroid aggregations of fibroin mass in the lumen and silk fibroin colloids, which were more evenly distributed (Fig. 4). Our findings demonstrated that the hydrophilic SER3 protein synthesized and secreted by PSG improves the water solubility and stability of the silk fibroin colloid in the lumen of SG, and further affects the polymerization and fibrogenesis of silk fibroin.

Studies have shown that the silk fibroin protein synthesized by silkworm PSG is present as fibroin units of Fib-H/Fib-L/P25 (molecular ratio 6:6:1) in silk fibers[9]. Among these proteins, P25 can form intermolecular interactions with Fib-H/Fib-L[7] and is evenly distributed in fibrils. Our results showed that in the silk fibers produced by the mutant, P25 broke away from the fibroin units and fibrils, and accumulated in the connecting layer between the silk core and the outer sericin (Fig. 2). We demonstrated that P25, a major component of the ancient cocoon silk structure, is substitutable, as also demonstrated by

a recent report of knocking out the P25 coding gene in *Bombyx mori*[8,33]. P25 is a glycoprotein containing Asn-linked oligosaccharide chains that forms a compact structure because of intramolecular disulfide linkages but associates with the H−L complex through non-covalent interactions[34]. Our recombinant protein SER3 had an NTD of Fib-H (Fig. 1c), which can associate with P25 by non-covalent interactions[35] and cause P25 to detach from the fibroin elementary units and then concentrated between the fibroin layer and the sericin layer for unknown reasons. However, SER's silk fibers exhibit greater advantages in deep processing than WT silk fibers (Fig. 3) and can prevent damage to the silk core fibrils in the degumming process. The silk fiber's textile material advantages remained unchanged, but new characteristics were additionally derived from the changes in the basic silk fibril structure.

Silk fibroin is a hydrophobic fibrous protein whose molecules are connected by disulfide bonds and whose secondary structure mainly comprises β-sheets[2,9,36]. SER3 protein is a hydrophilic globular protein whose secondary structure is dominated by random coils[37]. In the silk fibers produced by the mutant SER silkworms, the crystallization of the material increased, and the electron density in the crystalline and amorphous regions of the periodic structure changed, thus affecting the turning radius of the aperiodic structure in cocoon silk (Fig. 3). Correspondingly, the mechanical properties such as the maximum stress level and Young's modulus of SER silk fibers were also significantly improved, thereby enabling ultra-thin and ultra-dense fabrics to be woven (Supplementary Fig. 3). Moreover, the improved moisture absorption and liberation of the silk fibers, improved the performance of the textile material. Biocompatibility testing indicated that the fibroin fibers showed no adverse effects on the proliferation and growth of mammalian cells, thus indicating that SER silk fibroin had good biocompatibility, as compared with that of classical silk

fibroin (Supplementary Fig. 4). Our findings demonstrate the practical value of engineering applications.

## Efficiency of transgene-specific expression of foreign proteins in SGs of *Bombyx mori*

Since the piggyBac transposon-based expression system was developed in silkworms[38], dozens of transgenic silkworms with SG expression of foreign proteins have been established. However, the output of these foreign proteins is far lower than that of cocoon silk, and the higher the molecular weight of the foreign protein, the lower the output. Subsequently, researchers have made breakthroughs in increasing the expression levels of foreign proteins through continuous optimization. For example, with the *piggyBac*-mediated gene replacement system and transgenic technology, the spider's Major ampullate spidroin-1 gene (*MaSp1*) has been used to replace the silkworm *Fib-H* gene; after targeted integration into the PSG for expression, as much as 35.2% of the chimeric protein MaSp1 was obtained from cocoon silk fibers[22]. Related explorations have included the introduction of more than three foreign genes into the silkworm genome[39] and the use of enhancer combinations (hr3/IE1)[40]. Our laboratory has designed the artificial coding sequence *Hpl*, which is similar to *Fib-H*, and is specifically expressed in the PSG and binds Fib-L more strongly. In the cocoon silk produced by the transgenic silkworms, the content of the foreign protein HPL is 51.9% and 38.93% of the silk fibroin and cocoon silk, respectively[30]. Although these studies have significantly improved the expression efficiency of recombinant protein, they remain far from achieving the expression level of endogenous silk protein.

As shown in Supplementary Table 1, *Bombyx mori* expressed exogenous protein with molecular weight greater than 100 kDa in its silk glands, and were prone to silk gland development deformities, decreased survival, and a significantly diminished cocoon silk production efficiency, thus resulting in thin layered cocoon shells[24,28–30]. Although no description of abnormal cocoon silk yield has been provided in other reports, the expression of foreign proteins is generally not high (Supplementary Table 1). The highest content of foreign proteins reported is only 1.1% of the cocoon silk weight, and the expression in the posterior silk gland is less than 0.84% of the total cocoon silk[23,31,32]. The silk gland in silkworms is a highly specialized tissue with self-silk protein expression, and the expression of foreign proteins must be improved. The growth and development of the SGs and individual mutant silkworms in this study were normal. The weight of the cocoon shell, which reflects the protein synthesis and secretion function of the SG, exceeded that of the WT by 16.8%. The cocoon layer rate, which reflects the comprehensive production capacity of mature larvae, was 14.7% higher than that of the control (Supplementary Fig. 2). The content of SER3 protein in mutant cocoon silk was 4.3 times higher than that in the wild type, thus indicating that sericin SER3 in the posterior silk gland in mutants was expressed more efficiently than in the middle silk gland in the wild type (Fig. 2). We demonstrated that while suitable exogenous protein was expressed efficiently, the protein synthesis and secretion ability of by the silkworm SG were further improved.

In conclusion, we report an effective silkworm SG transgenic strategy. By selecting non-fibrous protein targets recombinantly expressed by the PSG, the metastable state of the silk protein, aqueous solution in the SG cavity was affected, thus enabling alteration of the composition, structure, and performance of the fibril molecules of the ancient silk fiber. The mutant completely overcame bottlenecks such as decreased viability, abnormal SG development, and low silk yield. Although the suitable SG transgene target proteins remain unclear, the results of this article provide a biological platform for effective indepth analysis of efficient specific silk protein synthesis by SG cells in the regulation of the synthesis of other proteins. This initial research may provide new ideas for bottom-up molecular design and biological production of silk protein materials.

## Methods

### Experimental animal preparation

The classic genetic strain N4W was used in this study. Larvae were reared on fresh mulberry leaves. The entire generation was maintained at 25.0 °C ± 2.0 °C in a natural light environment, except for special treatment methods. According to the steps described in Supplementary Text 1, the full-length sequence (3120 bp) of the sericin 3 gene (*Ser3*) specifically expressed in the MSG was cloned (Supplementary Sequence 1). The strategy in Fig. 1c and steps in Supplementary Text 1 were used to construct the *piggyBac* transgene vector and perform egg injection. The strategies and effects of mutant screening and genetic purification are shown in Supplementary Fig. 1, and the recombinant *Ser3* gene insertion site of the mutant was analyzed by tail-PCR sequencing (Supplementary Fig. 1g).

### Microscopic observation

The middle cocoon shell and degummed silk fiber of silkworm cocoons were observed by scanning electron microscopy (SEM). The spraying current was 20 mA, with platinum vacuum spraying for 3 min. Samples were observed by SEM (S4800, Hitachi, Japan) at room temperature, with repeated observations for three independent samples. The silk fiber was degummed and boiled in 0.2% $Na_2CO_3$ solution for 30 min. Meanwhile, the green fluorescence of EGFP was observed with a fluorescence microscope and laser confocal microscope to track the distribution of recombinant SER3 protein in cocoon silk fiber and silk gland tissue (section).

Transmission electron microscopy (TEM) was used to observe the raw silk samples and SG tissues. The samples were pre-cooled at 4 °C and fixed with electron microscope fixative (G1102, Servicebio, Wuhan, China) for 2 h, then fixed with 1% osmium acid for 2–4 h. The fixed samples were dehydrated with an ethanol gradient (50, 70, 80, 90, 95, and 100%) at 4 °C and then dehydrated with 100% ethanol and 100% acetone two times, with each dehydration lasting 15 min. After embedding and sectioning (thickness 60–80 nm), uranium-lead double staining (2% uranyl acetate saturated ethanol solution and lead citrate) was performed for 15 min each, and samples were dried at room temperature, then observed by TEM (HT7700, Hitachi, Japan).

### Silk fiber performance measurement

The mechanical properties were measured with a universal material testing machine (3365, Instron, USA) in a room with constant temperature and humidity (20 °C, R.H. 65%). The test conditions were as follows: initial length, 250 mm; tensile speed, 250 mm/min. A total of 20 WT or SER cocoons were boiled in water and fully expanded before reeling to obtain raw silk. The reeling cocoon number per raw silk was ten cocoons, and the reeling wire speed was 44–46 m/min. The obtained raw silk fiber retained most of the sericin, and one sample was taken every 3 meters, approximately between 100 and 200 meters, to determine the mechanical properties. A total of 22 samples were measured in the SER group, and 27 samples were measured in the WT group.

### Moisture absorption testing

After degumming, the silk fibers (n = 3 samples) were dried to a constant weight at 80 °C and then returned to normal temperature (20 °C) for accurate weighing. Hygroscopic properties were determined in a room with constant temperature and humidity (20 °C ± 2 °C, R.H. 65% ± 3%), and the weight was measured every 10 min until the fiber reached moisture absorption balance. When the moisture release performance was measured, the sample was first placed in an R.H. 100% container and sealed for 24 h, so that the fiber achieved moisture absorption balance (W0). Then the fibers were placed in a constant temperature, and humidity chamber (20 °C ± 2 °C, R.H. 65% ± 3%), and the quality changes were monitored continuously until the fiber reached a moisture balance. The regaining of moisture was expressed as the percentage of the mass of water absorbed or released by a unit

mass of silk fiber in different time periods with respect to the original fiber mass. The moisture absorption rate and moisture release rate are expressed as water mass absorbed or released by the silk fiber per unit mass at a certain time, according to a previously described method[41]. Origin2018 software was used to calculate the correlation constants of the fitting curve equation of regaining of moisture over time.

### Wide-angle X-ray diffraction (WAXD)

A D8 Advance X-ray diffractometer (Bruck, Germany) was used to identify the crystalline phase in the fibroin fibril samples. When scanning, the fibroin fibrils were cut into small pieces, placed on the sample stage, and scanned from 4 to 60° ($2\theta$) at a speed of 0.04°/s under the conditions of 40 kV and 40 mA (Cu target). The relative crystallinity of the sample was calculated in MDI jade 9 software. In the deconvolution process, the number and location of crystallization peaks were determined according to the data reported in the literature, and the peak area was calculated through baseline calibration, deconvolution, and peak fitting with the Pearson IV strategy. The crystallinity of the sample was calculated according to the following formula: crystallinity = (net area of diffraction peak/net area of diffraction peak + background area) × 100%.

### Small-angle X-ray scattering (SAXS)

The cocoon silk was measured by SAXS performed on (Nano-in Xider, Xenocs, France) with CuK$\alpha$ as a target. The samples of the cocoon shell in the middle layer were loaded into a porous sample rack at 25 °C and exposed for 200 s for individual measurements at a sample-to-detector distance of 938 mm. Scans were taken from 0 Å$^{-1}$ to 0.45 Å$^{-1}$ at a wavelength of 1.54 Å. Statistical analysis was performed in FIT2D and OriginPro 2022b software.

### Gene expression analysis

An RNAiso Plus (9109, TaKaRa, Dalian, China) was used to extract total RNA from PSG tissues of silkworm larvae on the third day of the 5th instar (5L3d). The cDNA was synthesized with a PrimerScript™ RT reagent kit with gDNA Eraser (Perfect Real Time) (RR047A, TaKaRa, Dalian, China) according to the manufacturer's instructions. qRT-PCR was performed in a total reaction volume of 20 μL with the TB Green® *Premix Ex Taq*™ (Tli RNaseH Plus) (RR420A, TaKaRa, Dalian, China), according to the manufacturers' instructions, and detected with ABI Stepone Plus (Ambion, Foster City, CA, USA). The *BmRp49* gene was selected as the internal control. Primers used in this study are listed in Supplementary Table 4.

### P25 immunofluorescence assays

Paraffin sections of cocoon silk fiber were made according to the conventional method. After dewaxing, sections were soaked in 0.01 M citrate buffer at 96 °C for 15 min for antigen repair, and the sections were exposed to a blocking solution for 40–60 min. P25 antibody was added to the tissue surfaces of the sections and incubated at room temperature for 1 h. The sections were washed three times with PBST for 5 min each. Then a TRITC-labeled secondary antibody (S0015, Affinity Biosciences, Ohio, USA) was added, incubated at room temperature for 1 h, then washed with PBST in the dark three times for 5 min each. Red fluorescence was observed with a fluorescence microscope (BX51, Olympus, Tokyo, Japan).

### Western blotting

A total of 0.5 g silk was added to 2 mL 9.3 M LiBr solution and completely dissolved at 60 °C for 4–6 h. The total protein concentration was measured with a BCA Protein Assay Kit (P0012, Beyotime, Shanghai, China). A 100 μg mass of total protein was electrophoresed by 10% SDS-PAGE and then transferred to a PVDF membrane. After blocking at 25 °C for 2 h, the primary antibodies, including rabbit anti-SER3 and rabbit anti-P25 (synthesized by Wuhan GeneCreate Biological Engineering Co., Ltd.), mouse anti-EGFP antibody (ab184601, abcam,

UK) and mouse anti-beta-Tubulin (ab108342, abcam, UK) was added to the membranes and incubated at 4 °C for 12 h, respectively. The membranes were washed three times with TBST. HRP-labeled goat anti-rabbit IgG or HRP-labeled goat anti-mouse IgG (Bioworld Technology, Minneapolis, MN, USA) was added and incubated at 37 °C for 2 h. Under dark conditions, 1 mL EZ-ECL chemiluminescence reagent was added to the membrane, and the bands were observed through chemiluminescence detection (1708370, Bio-Rad, USA) after 1 min. The images were analyzed with Image Lab (Bio-Rad, USA). The dilution of primary antibodies was 1:2000, and the dilution of secondary antibodies were 1:5000. All original blots are shown in source data.

### Statistics and reproducibility

Unless otherwise stated, each tissue sample was collected from at least three individuals, and each sample was measured three times. Each experiment was repeated three times independently with similar results. Image data are representative of three independent experiments unless otherwise stated. The data were statistically analyzed and graphically represented by GraphPad Prism (v8.0.2, GraphPad), and shown as mean ± standard error of mean (SEM), and the significance threshold was set at $p = 0.05$. The unpaired $t$ test analysis was used for the comparison only one group, and Holm–Sidak method was used for multiple $t$-test analysis for comparing three or more groups, and the $p$ value obtained was the adjusted p value.

### Reporting summary

Further information on research design is available in the Nature Research Reporting Summary linked to this article.

## Data availability

NCBI silkworm genome database (SilkDB, https://www.ncbi.nlm.nih.gov/genome/?term=Bombyx+mori) was used in the study. All data generated in this study are available within the article, Supplementary Information, and Source Data files. Source data are provided for Figs. 1–4 and Supplementary Figs. 1–4. Source data are provided with this paper.

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

## Acknowledgements

This work was supported by the National Natural Science Foundation of China (Grant No. 31972625 to S.X., 32102608 to Y.-F.W., and 31972873 to Y.-J.W.), the China Agriculture Research System of MOF and MARA (to Y.S.), the Priority Academic Program Development of Jiangsu Higher Education Institutions (PAPD to S.X.), Jiangsu Planned Projects for Postdoctoral Research Funds (2021K321C to Y.-F.W.), Nantong science and technology project (JC2021010 to Y.-F.W.). We acknowledge Anqi Liu and Xiaoning Sun for their help in the production of the schematic diagram of SGs producing cocoon silk.

## Author contributions

S.X., X.C., Y.-F.W., and Y.-J.W. conceived and designed the research. X.C., Y.-F.W., Y.-J.W., Q.L., X.L., J.L., R.P., and G.W. performed the experiments. S.X., Y.S., X.C., Y.-F.W., and Y.-J.W. reviewed the findings. S.X., Y.S., and Y.-F.W. contributed new reagents/analytic tools. S.X., X.C., Y.-F.W., and Y.-J.W. analyzed data. S.X., C.X., and Y.-F.W. wrote the initial manuscript. S.X., X.C., Y.-F.W., and Y.-J.W. performed the revision and editing of the manuscript. All authors have read and approved the manuscript.

## Competing interests

The authors declare no competing interests.

## Additional information

**Correspondence and requests** for materials should be addressed to Shiqing Xu.

