## [Peer Review File · Nature Communications]

Ectopic expression of sericin enables efficient production of ancient silk with structural changes in silkwormREVIEWER COMMENTS

Reviewer #1 (Remarks to the Author):

The manuscript entitled "Ectopic expression of sericin enables efficient production of ancient silk with structural changes in silkworm" by Chen et al., reported that a transgenic method was used in which the outer layer sericin SER3 in silk is secreted into the inner fibroin layer, thus generating a new structural fiber with non-fibrous sericin microsomes dispersed in fibroin fibrils.

My positive comments are as follows:

Silkworm silk is a super-long natural protein fiber with ancient structure pass thousands of years without change. The silk fiber produced by thousands of silkworm varieties around the world has almost the same composition, structure and characteristics. The authors implemented a new transgenic strategy to express water-soluble non-fibrin sericin in the posterior silk gland, thus altering the fibril structure and properties of the silk. The results provide new ideas for silk protein fiber molecular design.

I have no serious criticisms regarding results and figures. I am pleased to recommend publication if the authors could address the minor concerns listed below in a carefully revised form of this manuscript.

Specific comments:

1. How to understand the "ancient silk" used in the title and introduction? Does it mean that silk production has a long history, or that modern silk still retains the molecular structure of ancient silk, even since wild silkworm evolved into silkworm?
2. Silk gland has excellent ability of protein synthesis and secretion. This manuscript attempts to prove the reason why silk glands are often inefficient in expressing exogenous proteins, but the description is not sufficient and the explanation of this problem needs to be strengthened.
3. Is fibroin fibers the same as silk fibers? If yes, it must be consistent in the manuscript; if not, please explain. And what is the difference between silk fibroin and these two?
4. In Fig.3j, and Fig. S3, the picture of the Ultra dense fabrics and Ultra-thin fabrics are repeated. Delete one of them and revise the Figure.
5. In Figure S1, G0 adults is 85, G1 positive broods is 15, how is the mutation rate of 21% calculated?
6. The paper mentions: Figure S1 (c) transgenic mutants express red fluorescent protein RFP in the eye. What period is the eye of the silkworm? Need to explain!
7. What is content of the sericin SER3 of the transgenic silkworms in the cocoon silk produced?

Minor comments:

1. In the second paragraph on page 6, does "micro-body" in the sentence "Using immunofluorescence,...with different micro-body sizes (Fig.2d)" refer to "microsomes"? If yes, must be unify.
2. In the second paragraph on page 7, does "fibroin" in the sentence "The moisture absorption and desorption performance of fibroin showed significant improvements in the SER group" refers to fibroin fiber, or silk fibroin? Please clarify.
3. Desorption (in the second paragraph on page 7), dehumidification (in the second paragraph on page 7) and moisture liberation (in the abstract, and so on), do they mean the same? If yes, it must be consistent in the manuscript; if not, please explain.
4. In the last paragraph of the results section, MA, MP, PP and MM should be given their full names when they first appear in the manuscript.
5. In the sentence "Our laboratory has designed the artificial coding sequence Hpl, which is similar to Fib-H..." in paragraph 4 on page 11 of the discussion section, gene name of Hpl and Fib-H should be in italics.
6. Materials and Methods, incomplete source information of some reagents , such as lack of city and Cat number.
7. Figures legends of Fig.1, gene name should be in italics.
8. Supplementary Information, in Text 2, E. coli and S. aureus should not be abbreviated. This is the first appearance of these species.

Reviewer #2 (Remarks to the Author):

Innovative silkworm silk structures with higher fiber performance are in great demand. In the manuscript entitled “Ectopic expression of sericin enables efficient production of ancient silk with structural changes in silkworm”, the authors Chen et al. have ectopically expressed the outer layer sericin SER3 in the PSG of the silkworm by a piggyBac-mediated transgenic approach, thus generating a new fiber with improved β -sheet structure contents and mechanical properties, moisture absorption and moisture liberation properties. They found that the transgenic silkworm varieties have higher cocoon production efficiency without affecting silk gland development. Hence, they concluded it is an efficient, green method to produce new silk fibers with innovative properties *via* the silk gland transgenic target protein selection strategy.

The topic of this paper is very interesting. The authors have presented the observation and analysis after ectopic expression of SER3 in the PSG using TEM, FTIR etc. Several aspects of this paper - especially a more complete description and discussion of the method used and the results- could be improved including:

1) About novelty

Several previous studies have reported strategies to affect the mechanical properties of silk by overexpressing a specific protein (SER3) in the fibrin layer of silk fibers (see Refs. 23-28, provided by the authors). Although they state that SER3 is an endogenous protein specifically expressed in the MSG, but not in the PSG of the silkworm. SER3 should be considered as a foreign protein of the PSG. Therefore, it is not the first report on the use of exogenous proteins to modify the properties of silk fiber, and the results are predictable.

2) Insufficient analysis of mechanisms

1. In Fig. 2d, the authors have observed the separation of P25 from fibroin and the location of P25 between fibroin and sericin after expression of SER3 in the PSG. It is a very interesting result. How did SER3 change the component and structure of fibroin? The structure and properties of SER3 and whether it interacts with

Fib-H/Fib-L need to have a comparative analysis and a reasonable analysis of the experimental results is required.

2. As the authors have shown that SER3 has cysteine residue, why did SER3 not interact with Fib-H or Fib-L via disulfide bonds instead of an independent SM formation?
3. Why was P25 located between fibroin and sericin?
4. How did P25 affect the stability and structure of fibroin?
5. The authors have shown that SM is free or independent from fibroin. SER3 is predicted to have α -helix structure without β -sheet. How did SER3 expression result in an increase in β -sheet content of silk fiber?
6. Usually, the amorphous region formed by α -helix and random coil structure determines the elasticity and strain of animal silk. It is doubtful that the decrease in α -helix content will not affect the strain (Fig. 3c).
7. Line 256, "Moreover, the sericin microsomes dispersed in the fibrils significantly improved the moisture absorption and liberation of the silk fibers, thereby improving the performance of the textile material." In fact, the sericin microsomes were dispersed in the fibrils, which may disrupt the structure of mutant silk. How does it improve the mechanical properties of mutant silk?

3) Technical and methodological issues

1. In Fig. 1b/2b, more evidence such as MS, WB should be provided to claim SM observed in the fibrils.
2. In Fig. 2a, an image of silk fiber under white light should be provided to clearly indicate the details of silk fiber.
3. In Fig. 2c, it is undoubtful that the sericin content in silk layer increased as SER3 was overexpressed in the PSG and then secreted into silk fiber. The authors should measure the expression of SER3 in the PSG and the content of SER3, Fib-H, Fib-L and P25 in the silk fiber to better support the claim of high silk yield. However, we also noticed that there is no significant difference in cocoon weight (Fig. S2i) and an increase in the cocoon layer ratio (Fig. S2j), which indicated a

decrease in the pupal weight of SER3 silkworm. Hence, the high silk yield via ectopic expression of SER3 in the PSG is not convincing.

4. In Fig. 2c, the authors stated that the percentage of sericin in cocoon silk in the SER group was 7.39% higher than that in the WT group, an increase in 21.8%. How did the authors calculate the percentage of sericin in cocoon in SER3 group and make the comparison with WT group? I suggest they perform SDS-PAGE and western blot to clearly show the expression of SER3 in the fibroin layer, but not the actual yield of silk protein.
5. In Fig. 2e-f, the result is not convincing as different peak numbers were applied for the curve-fitting and the determination of the secondary structure. Also, the same wavenumber was assigned to different secondary structure. In fact, according to the data provided by the authors, the curve-fitting results using a general procedure showed that the β -sheet content was 42.05% in the WT group and 44.38% in the SER3 group. The difference between the two groups did not appear to be significant.
6. The authors should provide more evidence that the silk structure has been indeed changed. I suggest to perform SAXS or WAXS for characterizing silk crystal structure and size.
7. As shown in Fig. 4c, it is strange that Fib-H/L/P25 appeared in MP. I suggest the authors should take care the part of silk gland for PCR. Also, there is a significant difference between the relative expression of EGFP and SER3. Could the authors explain the difference in the expression of EGFP and SER3 since they are fused together?
8. According to the results provided by the authors, the silk structure of the mutant was changed dramatically. Then the author should analyze the microstructure of mutant silk more comprehensively, as FTIR is usually considered as a semi-quantitative method. Wide-angle X-ray diffraction or small-angle X-ray scattering could clearly characterize the crystallinity, grain size, orientation and other key structural information of polymer materials, which should be supplemented to better indicate the structural changes of mutant silk.

9. It is suggested that proteomic analysis of degumming mutant silk should be performed to precisely determine SER3 content in silk fiber, which is essential to address the mechanism by which SER3 affects the structure and properties of silk.
10. The author mentioned that the number of cocoons for tensile testing was 20. In the data provided by the authors, the number of samples was 22 (SER) and 27 (WT). It can be assumed that only 1 or 2 silks strands from each cocoon are used for testing. Due to the obvious variance in the mechanical properties of silk, this method does not seem to accurately reflect the overall silk properties.
11. Generally, a length of 100 mm and a tensile speed of 100 mm/min are applied for mechanical performance test. It is obvious the length and tensile speed may affect the mechanical performance of silk fiber. Could the authors please explain why did you perform the test with the parameters different from literatures?
12. The diameter and cross-sectional area of the silk may affect the stress. Hence, the authors should provide the diameter and the cross-sectional area of silk and indicate how the cross-sectional area is determined.

4) writing issues

1. The abstract should be well revised to better indicate the most important findings and the significance of this study. For example, the outer layer sericin SER3 was ectopically expressed in the PSG of the silkworm via a piggyBac-mediated transgenic approach, then secreted into the inner fibroin layer, thus generating a new fiber with sericin microsomes dispersed in fibroin fibrils. The cause and effect in the sentence “Moreover, the water solubility and stability of the fibroin-colloid in the silk glandular cavity are increased, thus significantly improving the β -sheet content of fibroin, as well as the mechanical properties, moisture absorption and moisture liberation of the silk fiber” is not valid.
2. The introduction should be well revised to clearly indicated the purpose, contents and significance of this study. I’m confused about the mechanism of the metastability of ultra-high concentration aqueous solutions of Fib-H/Fib-L/P25 polymers in SGs, or altering the ancient silk structure via innovative

reprogramming of the genomes of SG cells with high survival rate and silk yield. It is difficult to understand the relationship between the sentences “the fibril structure and function of the ancient silk fiber were greatly altered” and “This method may help address the bottleneck problems of the low survival rate and low silk yield of genetically transgenic silkworms”. Also, I’m confused that the function of the ancient silk fiber was greatly altered. What is the function of the ancient silk fiber and how the function of the fiber was changed?

3. In Fig. 1a, spinning dope may be redundantly labeled.
4. In Fig. 1c, PiggBac should be piggyBac.
5. The gene names in the Fig. 1c, Figures legends (line 489–496), and Materials and Methods (line 300) are not italics; the mutant strain name, gene name and protein name (SER, SER I, SER II, SER III, SER3, ser and ser3) are confused in the manuscript.
6. In Fig. 2a, the abbreviations of fibroin layer, SF and F should be unified.
7. In Fig. 3, The sample was a cocoon silk fiber (100–200 meters) without the sericin protein of the outer layer removed (n=20 cocoons). I am confused about the presence or absence of sericin protein.
8. Fig.3 k-n are not closely related with the mechanical performance of silk fiber and should be removed or stated in the supplementary materials.
9. Fig. 4a should be well re-organized to clearly indicate the different part of silk gland.
10. Fig. S2, figure and figure legend of c, d, e and f are not appropriate.
11. Fig. S3c, dvunitka → ultra-dense?
12. Line 366, μl → μL
13. Line 65-67, the author mentioned the reprogramming of the genomes of SG cells, which is irrelevant to the topic of this study and should therefore be deleted.
14. The manuscript should be well proof edited by a native English speaker to polish the grammar, expression and organization and correct the typos.

Reviewer #3 (Remarks to the Author):

This manuscript describes the production of a novel type of *B. mori* silk fiber with different molecular compositions and fiber morphology from normal silk using a transgenic technology. The authors showed that such differences have led to better mechanical and moisture absorption properties of silk. The effects of transgenesis on silk's properties are unique and might be useful in practical applications. Therefore, this manuscript would have considerable impacts on the researchers in the field of proteins materials. However, some of the authors' conclusions are not fully supported by the data in the manuscript. This manuscript is thus not appropriate for publication in *Nature Communications* in the present form.

Major issues:

(1) Line 72-79.

The authors describe many negative aspects of previous genetic alterations of silkworms. Although there are many successful examples of genetic alterations, they seem to emphasize negative aspects of genetic engineering too much by introducing some specific examples such as expressing "cytotoxin" in PSG (ref 29). The authors should summarize previous studies in a fair position.

(2) Line 82-83.

This sentence would cause misunderstanding that the present genetic engineering method is not suitable for practical applications. There are actually some trials for commercial productions of genetically engineered silks in some countries. It is also unclear how the method proposed in this manuscript can solve the bottlenecks shown here. Therefore, this sentence should be omitted or revised. Moreover, the meaning of "low survival rate" is unclear, and "low silk yield" is not observed in many cases except for some specific examples such as ref 29.

(3) Figure 1b.

Sericin II (may be equal to Ser2) is reported to be major coating proteins of larval silk threads spun during the growing stages (Takasu et al. *Insect Biochem Mol Biol.* 2010 Apr;40(4):339-44. doi: 10.1016/j.ibmb.2010.02.010). In addition, I am not sure that the layered structure of sericins has been experimentally verified so far. Please add some appropriate references for this model.

(4) Line 99-101.

The presence of sericin 3 protein in the microsomes should be verified by immunostaining or fluorescence observation of fused GFP.

(5) Line 101-103.

The authors describe that "the production efficiency of cocoon silk was significantly higher than that of the wild type (WT)". But no data is shown here. Figure S2i and j respectively show total cocoon weights and the ratio of silk in the total cocoon weights including pupa. The authors should clearly show the comparison of silk and silk fibroin production between SER and WT.

(6) Line 105-106.

The description "the transgenic silkworm SGs have superior production performance" is not supported by experimental data because no data on silk production is provided.

(7) Line 118-119.

The authors show the percentage of sericin in silk. However, without the amounts of silk and silk fibroin, it is impossible to know whether the increase of the percentage resulted from the increase of sericin production or the decrease of fibroin production.

(8) Line 119-120.

This sentence is not supported without the data of actual amounts of total silk, fibroin, and sericin. SDS-PAGE analysis is also required to show the increase of Ser3 production.

(9) Line 161-162.

Some quantitative analyses such as molecular weight analysis by SDS-PAGE or GPC are required to conclude that the mutant silk has higher alkali resistance than the WT silk.

(10) Line 216.

Since the authors showed only one example of a mutant silk with changed properties in this manuscript, the word “controllable” is not appropriate here. Please delete it.

(11) Figure 2e.

Since the number of the deconvoluted peaks are different among samples, direct comparison might be inappropriate. More careful discussion is necessary for structural analysis. The assignment of the left-most peaks is different among samples. Please explain why.

(12) Line 250.

To discuss the change of the structure of silk fibroin, using only an IR analysis is not sufficient. The interpretation of the IR spectra is not convincing as described in (12). It is preferable to combine with other methods such as X-ray analysis. If the discussion of structural changes is not essential for the conclusions, I recommend that the IR analysis results are omitted.

(13) It is preferable to show the data of silk obtained from heterozygous individuals (hybrid of SER and WT lines). If such data is intermediate between SER and WT, the conclusions will be more convincing.

Minor issues:

(14) The word “piggyBac” should be written in italics.

(15) Unify the abbreviations of silk fibroin and sericin in Figure 2 (“SF”, “F”, “Fibroin”, “SS”, and “S” are mixed and thus confusing).

(16) The words “TALEN” in Line 302 in the main text and Line36 in the SI might be mistyping of “piggyBac”.

REVIEWER COMMENTS

Reviewer #1

The manuscript entitled "Ectopic expression of sericin enables efficient production of ancient silk with structural changes in silkworm" by Chen et al., reported that a transgenic method was used in which the outer layer sericin SER3 in silk is secreted into the inner fibroin layer, thus generating a new structural fiber with non-fibrous sericin microsomes dispersed in fibroin fibrils.

My positive comments are as follows:

Silkworm silk is a super-long natural protein fiber with ancient structure pass thousands of years without change. The silk fiber produced by thousands of silkworm varieties around the world has almost the same composition, structure and characteristics. The authors implemented a new transgenic strategy to express water-soluble non-fibrin sericin in the posterior silk gland, thus altering the fibril structure and properties of the silk. The results provide new ideas for silk protein fiber molecular design.

I have no serious criticisms regarding results and figures. I am pleased to recommend publication if the authors could address the minor concerns listed below in a carefully revised form of this manuscript.

Response: Thank you very much for reviewing our manuscript and giving such a positive opinion. Your comments are all valuable and very helpful for revising and improving our manuscript, as well as the important guiding significance to our researches. We have tried our best to improve the manuscript and have made a lot of changes which we hope meet with approval.

Major comments:

Major comment 1: How to understand the "ancient silk" used in the title and introduction? Does it mean that silk production has a long history, or that modern silk still retains the molecular structure of ancient silk, even since wild silkworm evolved into silkworm?

Response: Yes, the modern silk still retains the molecular structure of ancient silk, although it has been thousands of years since silkworm was used to produce silk. We use the word 'ancient silk' by borrowing the usage of Omenetto and Kaplan (Science, 2010) [1] on silkworm cocoon silk.

The silk fiber of silkworm cocoon has a core-shell type structure, with silk fibroin as the inner core and sericin as the outer coating. Each silk fibroin brin is composed of numerous interlocking fibroin fibrils [2]. Fibroin is the main component, accounts for > 70% of cocoon silk proteins, and is composed of Fib-H, Fib-L, and P25 proteins in a 6:6:1 molar ratio [3-5]. At present, more than a thousand silkworm varieties have been selected and bred, but modern silk still retain the molecular structure of wild silkworm cocoon silk.

References

- [1] Omenetto FG, Kaplan DL. New opportunities for an ancient material. *Science*. 2010, 329(5991):528-531.
- [2] Huang W, et al. Silkworm silk-based materials and devices generated using biotechnology. *Chem Soc Rev*. 2018, 47(17):6486-6504.
- [3] Li G, et al. Silk-based biomaterials in biomedical textiles and fiber-based implants. *Adv Healthc Mater*. 2015, 4(8):1134-1151.
- [4] Hao Z, et al. New insight into the mechanism of *in vivo* fibroin self-assembly and secretion in the silkworm, *Bombyx mori*. *Int J Biol Macromol*. 2021, 169:473-479.
- [5] Inoue S, et al. Silk fibroin of *Bombyx mori* is secreted, assembling a high molecular mass elementary unit consisting of H-chain, L-chain, and P25, with a 6:6:1 molar ratio. *J Biol Chem*. 2000, 275(51):40517-40528.
- [6] Xiang H, et al. The evolutionary road from wild moth to domestic silkworm. *Nat Ecol Evol*. 2018, 2(8):1268-1279.

Major comment 2: Silk gland has excellent ability of protein synthesis and secretion. This manuscript attempts to prove the reason why silk glands are often inefficient in expressing exogenous proteins, but the description is not sufficient and the explanation of this problem needs to be strengthened.

Response: Thank you for your constructive suggestion. Relevant descriptions have been added to the discussion of the revised version.

Original text: No ideal solution has been described to address the bottleneck problems that commonly occur in SG target tissue transgenic silkworms, such as reduced viability, abnormal SG development and low silk yield^{29,30}. The growth and development of the SGs and individual mutant silkworms in this study were normal. The weight of the cocoon shell, which reflects the protein synthesis and secretion function of the SG, exceeded that of the WT by 16.8%. The cocoon layer rate, which reflects the comprehensive production capacity of mature larvae, was 14.7% higher than that of the control (Supplementary Fig. 2). We demonstrated that while suitable exogenous protein was expressed efficiently, the protein synthesis and secretion ability of the silkworm SG were further improved.

Revision: As shown in Table S1, *Bombyx mori* expressed exogenous protein with molecular weight greater than 100 kDa in its silk gland (SG), which was prone to silk gland development deformity and decreased individual survival rate, and the cocoon silk production efficiency was significantly reduced, resulting in thin layered cocoon shells^{24,28-30}. Although there is no description of abnormal cocoon silk yield in other reports, the expression of foreign proteins is generally not high. The highest content of foreign proteins reported is only 1.1% of the cocoon silk weight, of which the expression in the posterior silk gland is less than 0.84% of the cocoon silk [5-7]. It shows that the silk gland of silkworm is a highly specialized self-silk protein expression tissue, and the function of expressing foreign proteins needs to be improved.

The growth and development of the silkworm and SG of mutants in this study were normal. The weight of the cocoon shell, which reflects the protein synthesis and secretion function of the SG, exceeded that of the WT by 16.8%. The cocoon layer rate, which reflects the comprehensive production capacity of mature larvae, was 14.7% higher than that of the control (Fig. S2). The content of SER 3 protein in mutant cocoon silk was 4.3 times higher than that in the wild type, thus indicating that sericin SER 3 in the posterior silk gland in mutants was expressed more efficiently than in the middle

silk gland in the wild type (Fig. 2). We demonstrated that while suitable exogenous protein was expressed efficiently, the protein synthesis and secretion ability of by the silkworm SG were further improved.

References

- [1] Otsuki, R. et al., Bioengineered silkworms with butterfly cytotoxin-modified silk glands produce sericin cocoons with a utility for a new biomaterial. *Proc. Natl. Acad. Sci. USA* 114, 6740-6745 (2017).
- [2] Minagawa, S. et al. Production of a correctly assembled fibrinogen using transgenic silkworms. *Transgenic Res* 29, 339-353 (2020).
- [3] Wang, H. et al. High yield exogenous protein HPL production in the Bombyx mori silk gland provides novel insight into recombinant expression systems. *Sci. Rep.* 2015, 5:13839.
- [4] Teulé, F. et al. Silkworms transformed with chimeric silkworm/spider silk genes spin composite silk fibers with improved mechanical properties. *Proc. Natl. Acad. Sci. USA* 109, 923-928 (2012)
- [5] Kuwana, Y. et al. High-toughness silk produced by a transgenic silkworm expressing spider (*Araneus ventricosus*) dragline silk protein. *PLoS One* 9, e105325 (2014)
- [6] Iizuka, M. et al. Production of a recombinant mouse monoclonal antibody in transgenic silkworm cocoons. *FEBS. J* 276, 5806-5820 (2009)
- [7] Tomita, M. et al. Transgenic silkworms produce recombinant human type III procollagen in cocoons. *Nat. Biotechnol* 21, 52-56 (2003).

Major comment 3: Are fibroin fibers the same as silk fibers? If yes, it must be consistent in the manuscript; if not, please explain. And what is the difference between silk fibroin and these two?

Response: The fibroin fibers are different from silk fibers. The fibroin fibers refer to the fiberized fibroin component inside the silk fibers.

As shown in Figure R1, silk fiber is mainly composed of sericin in the outer layer and fibroin in the inner layer. The sericin is highly hydrophilic, which acts as an adhesive joining two fibroin filaments in order to form cocoon silk, which is also known as silk fibers. Hydrophobic fibroin has good mechanical properties and is converted into raw silk and used in the production of many types of yarns and silk fabrics.

Figure R1. Hierarchical structure of *B. mori* silk [1]. (a) *B. mori* silk fiber has a core-shell type structure, with silk fibroin as the inner core and sericin as the outer coating. Each silk fibroin brin is composed of numerous interlocking fibroin fibrils. Inside the fibroin fibrils, the β -sheet nanocrystals are connected by amorphous chains to form a heteronanocomposite. β -sheet nanocrystals are composed of stacked β -sheets with peptide chains connected by hydrogen bonds in each sheet. The lattice constants of the orthogonal unit cell of β -sheet nanocrystal are $a = 0.938$ nm, $b = 0.949$ nm, and $c = 0.698$ nm for silkworm silk. (b) Scanning electron microscopy image of native *B. mori* silkworm silk. (c) Atomic force microscopy image of the fibroin fibril structure in *B. mori* silkworm silk with a sequence of linked segments.

Editorial note: Used with permission of the Royal Society of Chemistry, from Silkworm silk-based materials and devices generated using bio-nanotechnology. Huang, Wenwen; Kaplan, David L.; Li, Chunmei; Ling, Shengjie; Omenetto, Fiorenzo G., 47, 17, 2018; permission conveyed through Copyright Clearance Center, Inc.

References

[1] Huang W, et al. Silkworm silk-based materials and devices generated using bio-nanotechnology. *Chem Soc Rev.* 2018, 47(17):6486-6504.

Major comment 4: In Figure 3j, and Figure S3, the picture of the Ultra dense fabrics and Ultra-thin fabrics are repeated. Delete one of them and revise the Figure.

Response: Thank you for your correction. We have revised in the revision. **Major**

comment 5: In Figure S1, G0 adults is 85, G1 positive broods is 15, how is the mutation rate of 21% calculated?

Response: We have revised the **Fig. S1a** in the revision. The total number of microinjected eggs was 1820, and 580 larvae (G0 generation) were incubated, with a hatching rate of 32%. A total of 85 adults were obtained from G0 generation larvae, of which 71 obtained offspring (G1 generation), and the remaining 14 did not obtain offspring. Among the 71 G1 broods, positive broods were 15, and the positive rate is 21% (**Fig. S1a**).

Original Fig. S1a

Target gene	Injected eggs	Hatched (%)	G0 adults	Mutant (%)	G1 positive broods
BmSer3	1820	32	85	21	15

Revised Fig. S1a

Target gene	Injected eggs	Hatched (%)	G0 adults	G1 broods	Positive broods	Mutant (%)
BmSer3	1820	32	85	71	15	21

Major comment 6: The paper mentions: Figure S1 (c) transgenic mutants express red fluorescent protein RFP in the eye. What period is the eye of the silkworm? Need to explain!

Response: The detection period has been marked in the Revised legend of **Fig. S1**. We use the silkworm neural specific 3×P3 promoter to regulate *RFP* reporter gene, which can make the silkworm eye show specific red fluorescence at 554 nm excitation wavelength. Therefore, the eyes of SER transgenic silkworm positive individuals can

show specific red fluorescence at 554 nm excitation wavelength at the whole stages of

larva, pupa and adult, as well as the end of embryonic development. In Fig. S1c, we detected the red fluorescence of eyes at the 3rd day of the 5th instar larvae.

Major comment 7: What is content of the sericin SER3 of the transgenic silkworms in the cocoon silk produced?

Response: Using a classical degumming method of cocoon silk to determine the sericin content in cocoon silk, the percentage of sericin in the SER group was 40.77%, and was 7.39% higher than that in the WT group (Revised Figure 2e).

In the revision, the western blotting was used to determine the SER3 content in cocoon silk with P25 as internal reference. The results showed that the SER3 content in mutant was 4.3 times higher than that in WT group (Revised Figure 2f).

Minor comments:

1. In the second paragraph on page 6, does "micro-body" in the sentence "Using immunofluorescence, ... with different micro-body sizes (Figure 2d)" refer to "microsomes"? If yes, must be unify.

Response: Accepted. We have revised in the revision.

2. In the second paragraph on page 7, does "fibroin" in the sentence "The moisture absorption and desorption performance of fibroin showed significant improvements in the SER group" refers to fibroin fiber, or silk fibroin? Please clarify.

Response: The "fibroin" in the sentence refers to fibroin fiber, which is silk fiber with sericin removed. We have revised in the revision.

3. Desorption (in the second paragraph on page 7), dehumidification (in the second paragraph on page 7) and moisture liberation (in the abstract, and so on), do they mean the same? If yes, it must be consistent in the manuscript; if not, please explain.

Response: Thanks for the corrections. Desorption, dehumidification and moisture liberation express the same meaning in the manuscript, which has been unified as moisture liberation. We have revised in the revision.

4. In the last paragraph of the results section, MA, MP, PP and MM should be given their full names when they first appear in the manuscript.

Response: Accepted. MA, MM and MP show the anterior, middle and posterior parts of the middle silk gland, respectively. PA and PP show the anterior and posterior parts of the posterior silk gland, respectively. We have revised in the revision.

5. In the sentence "Our laboratory has designed the artificial coding sequence Hpl, which is similar to Fib-H..." in paragraph 4 on page 11 of the discussion section, gene name of Hpl and Fib-H should be in italics.

Response: Accepted. We have revised in the revision.

6. Materials and Methods, incomplete source information of some reagents, such as lack of city and Cat number.

Response: Accepted. We have added the information in the revision.

7. Figures legends of Figure 1, gene name should be in italics.

Response: Accepted. We have revised in the revision.

8. Supplementary Information, in Text 2, *E. coli* and *S. aureus* should not be abbreviated. This is the first appearance of these species.

Response: Accepted. We have added the full name of *Escherichia coli* (*E. coli*) and *Staphylococcus aureus* (*S. aureus*) in the revision.

Reviewer #2

Innovative silkworm silk structures with higher fiber performance are in great demand. In the manuscript entitled ‘Ectopic expression of sericin enables efficient production of ancient silk with structural changes in silkworm’, the authors Chen et al. have ectopically expressed the outer layer sericin SER3 in the PSG of the silkworm by a piggyBac-mediated transgenic approach, thus generating a new fiber with improved β -sheet structure contents and mechanical properties, moisture absorption and moisture liberation properties. They found that the transgenic silkworm varieties have higher cocoon production efficiency without affecting silk gland development. Hence, they concluded it is an efficient, green method to produce new silk fibers with innovative properties via the silk gland transgenic target protein selection strategy.

The topic of this paper is very interesting. The authors have presented the observation and analysis after ectopic expression of SER3 in the PSG using TEM, FTIR etc. Several aspects of this paper - especially a more complete description and discussion of the method used and the results- could be improved including:

Response: Thank you very much for reviewing our manuscript and giving such a detailed opinion. Your comments are all valuable and very helpful for revising and improving our manuscript, as well as the important guiding significance to our researches. We have tried our best to improve the manuscript and have made a lot of changes which we hope meet with approval.

Major comment 1: Several previous studies have reported strategies to affect the mechanical properties of silk by overexpressing a specific protein (SER3) in the fibrin layer of silk fibers (see Refs. 23-28, provided by the authors). Although they state that SER3 is an endogenous protein specifically expressed in the MSG, but not in the PSG of the silkworm. SER3 should be considered as a foreign protein of the PSG. Therefore, it is not the first report on the use of exogenous proteins to modify the properties of silk fiber, and the results are predictable.

Response: As you pointed out, it is not the first report on the use of exogenous proteins to modify the properties of silk fiber. However, there is no report of using sericin 3 (*Ser3*). The Refs. 23-28 reported the exogenous proteins expressed in the silk glands:

23. Song, Z. et al. Reducing blood glucose levels in T1DM mice with an orally administered extract of sericin from hIGF-I-transgenic silkworm cocoons. *Food. Chem. Toxicol* 67, 249-254 (2014): A human insulin-like growth factor-I (hIGF-I) gene controlled by the sericin-1 promoter with the signal peptide DNA sequence of the fibrin heavy chain (*Fib-H*). The recombinant hIGF-I is synthesized and excreted by middle silk gland and transferred to the sericin layer of cocoon silk.
24. Li, Z. et al. Construction of transgenic silkworm spinning antibacterial silk with fluorescence. *Mol. Biol. Rep* 42, 19-25 (2014): A targeting vector consisting of a fusion gene of the green fluorescent protein gene *gfp* and the antimicrobial peptide cecropin gene *cec* flanked by pieces of the 5' and 3' sequences of the *Fib-L* of the silkworm and a negative selection DsRed marker gene driven by the baculovirus immediate early gene 1 promoter, was used to target the silkworm genome in order to explore the possibility of improving the antibacterial properties of silk.

25. Wang, F. et al. An optimized sericin-1 expression system for mass-producing recombinant proteins in the middle silk glands of transgenic silkworms. *References Transgenic. Res* 22, 925-938 (2013): Transgenic middle silk gland (MSG) expression systems based on the usage of promoter of sericin1 gene to produce recombinant proteins in MSG. They provide an alternative modification strategy by using the hr3 enhancer (hr3 CQ) from BmNPV virus and the 3'UTRs of Fib-H, Fib-L, and Sericin1 genes.
26. Iizuka, T. et al. Colored fluorescent silk made by transgenic silkworms. *Adv. Funct. Mater* 23, 5232-5239 (2013): Using promoter of Fib-H gene to produce recombinant proteins of EGFP, DsRedMonomer and mKO in posterior silk gland (PSG), then made green, red and orange colored fluorescent cocoon silk.
27. Xue, R. et al. Expression of hGM-CSF in silk glands of transgenic silkworms using gene targeting vector. *Transgenic. Res* 21, 101-111 (2012): To express human granulocyte-macrophage colony-stimulating factor (hGM-CSF) in the posterior silk glands of silkworms, a targeting vector pSK-FibL-L-A3GFP-PH-GMCSF-LPA-FibL-R was constructed, harboring a 1.2 kb portion of the left homogenous arm (FibL-L), a 0.5 kb portion of the right homogenous arm (FibL-R), fibroin H-chain-promoter-driven hGM-CSF and silkworm actin 3-promoter-driven GFP.
28. Teulé, F. et al. Silkworms transformed with chimeric silkworm/spider silk genes spin composite silk fibers with improved mechanical properties. *Proc. Natl. Acad. Sci. USA* 109, 923-928 (2012): Using promoter of silkworm Fib-H gene, its upstream enhancer element, its N-terminal domain and C-terminal domain, and a marking gene EGFP, a part of spider silk sequence was expressed in the posterior silk glands of silkworm.

Although the efforts to express and secrete exogenous proteins in the SGs of silkworms through transgenic technology to date have many successful examples of genetic alterations. However, it is still a great challenge to greatly improve the expression efficiency of foreign proteins while maintaining the cocoon silk yield, especially to express high molecular weight proteins (~100 kDa) in the posterior silk glands [1-7].

As shown in Table S1, *Bombyx mori* expressed exogenous protein with molecular weight greater than 100 kDa in its silk glands, and were prone to silk gland development deformities, decreased survival, and a significantly diminished cocoon silk production efficiency, thus resulting in thin layered cocoon shells [1-4]. Although no description of abnormal cocoon silk yield has been provided in other reports, the expression of foreign proteins is generally not high (Table S1). The highest content of foreign proteins reported is only 1.1% of the cocoon silk weight, and the expression in the posterior silk gland is less than 0.84% of the total cocoon silk [5-7]. The silk gland in silkworms is a highly specialized tissue with self-silk protein expression, and the expression of foreign proteins must be improved.

In this manuscript, a high molecular weight foreign protein (recombinant SER3 protein, 142 kDa) is expressed in the posterior silk gland to modify silk fiber. The highlights:

(1) No adverse effect has been observed on the vitality of transgenic mutant silkworm, the survival of SER larvae infected with bacteria under stress was higher than that of the WT (revised Fig. S2m & S2n). Furthermore, no developmental

phenotypic differences were observed in the silk glands of the 5th instar larvae (revised Fig. S2f).

(2) The cocoon layer weight of SER was 116.8% of that of WT group, from 0.104 g per cocoon in WT group to 0.123 g in SER group (revised Fig. S2k). The cocoon layer rate (cocoon silk production efficiency) of the SER silkworm was 114.8% of that of the WT, from 10.64% in WT group to 12.22% in SER group (revised Fig. S2l). The content of SER3 in the mutant cocoon silk (including SER3 expressed in the middle silk gland) was 4.3 times higher than that of the WT (revised Fig. 2f).

(3) The cocoon silk fiber of mutant showed significant changes in the molecules composition and structure of fibroin, such as (A) generating a new structural fiber with non-fibrous sericin microsomes dispersed in fibroin fibrils; (B) the P25 detached from the fibroin unit of Fib-H/Fib-L/P25 polymer, and accumulated on the surface of fibroin fiber.

References

- [1] Otsuki, R. et al., Bioengineered silkworms with butterfly cytotoxin-modified silk glands produce sericin cocoons with a utility for a new biomaterial. *Proc. Natl. Acad. Sci. USA* 114, 6740-6745 (2017).
- [2] Minagawa, S. et al. Production of a correctly assembled fibrinogen using transgenic silkworms. *Transgenic Res* 29, 339-353 (2020).
- [3] Wang H, et al. High yield exogenous protein HPL production in the *Bombyx mori* silk gland provides novel insight into recombinant expression systems. *Sci Rep.* 2015, 5:13839.
- [4] Teulé, F. et al. Silkworms transformed with chimeric silkworm/spider silk genes spin composite silk fibers with improved mechanical properties. *Proc. Natl. Acad. Sci. USA* 109, 923-928 (2012)
- [5] Kuwana, Y. et al. High-toughness silk produced by a transgenic silkworm expressing spider (*Araneus ventricosus*) dragline silk protein. *PLoS One* 9, e105325 (2014)
- [6] Iizuka, M. et al. Production of a recombinant mouse monoclonal antibody in transgenic silkworm cocoons. *FEBS. J* 276, 5806-5820 (2009)
- [7] Tomita, M. et al. Transgenic silkworms produce recombinant human type III procollagen in cocoons. *Nat. Biotechnol* 21, 52-56 (2003).

Major comment 2: In Figure 2d, the authors have observed the separation of P25 from fibroin and the location of P25 between fibroin and sericin after expression of SER3 in the PSG. It is a very interesting result. How did SER3 change the component and structure of fibroin? The structure and properties of SER3 and whether it interacts with Fib-H/Fib-L need to have a comparative analysis and a reasonable analysis of the experimental results is required.

Response: Thank you for your constructive suggestion. The related analysis has been added to the appropriate position in the revised manuscript.

As shown in Figure.R2 and Fig.1a, silk fibroin is secreted from PSG as a 2.3-MDa elementary unit, consisting of six sets of a disulfide-linked heavy chain (Fib-H)-light chain (Fib-L) heterodimer and one molecule of fibrohexamerin (P25) [1, 2]. The H-L dimer forms a disulfide bond between Cys172 in Fib-L and the twentieth residue from the carboxyl terminus of Fib-H (Cys-c20) [3]. One internal P25 protein and six H-L

dimers form the fibroin elementary unit on the cell endoplasmic reticulum through non-covalent interactions with the NTD of Fib-H [4]. The current research shows that the deletion of Fib-H and Fib-L will lead to the inability of silk gland to secrete fibroin [5-7], while the deletion of P25 will not affect the formation of fibroin fiber [8], indicating that P25 is not necessary to form the basic unit of fibroin. Silk fiber is composed of sericin in the outer layer and fibroin in the inner layer. The sericin is highly hydrophilic, which is expressed in the middle silk gland (MSG), while fibroin is highly hydrophobic and expressed in the posterior silk gland (PSG) [9, 10].

The ratio of the number of cysteine molecules in the amino acid residues of the recombinant SER3 protein (0.50 %) was intermediate between that of Fib-H (0.10 %) and Fib-L (1.10 %) (Table S2), which has the possibility form disulfide bonds and combine with Fib-H and other silk fibroin in the PSG. As shown in Fig. 4a, recombinant SER3 protein expressed in the PSG of the mutant is unevenly distributed in the fibroin, thus indicating that the recombinant SER3 cannot form disulfide bonds and combined to Fib-H and Fib-L in the PSG. When the silk protein of PSG enters MSG, some hydrophilic SER3 will disperse from the fibroin layer and integrate into sericin layer. Meanwhile, recombinant SER3 protein can be connected to P25 with their NTD, this NTD is same as the Fib-H (Fig.1c). Thus, P25 protein may be partially separated along with recombinant SER3 protein from the interior of fibroin and distributed to the outer layer.

Figure R2. Model describing the structural hierarchy and molecular assembly in *B. mori* cocoon silk [1].

Editorial note: Reprinted (adapted) with permission from Peng, Z., Yang, X., Liu, C. et al. Structural and mechanical properties of silk from different instars of *Bombyx mori*. *Biomacromolecules*. 2019, 20(3):1203-1216. Copyright 2019 American Chemical Society."

References"

- [1] Peng Z, et al. Structural and mechanical properties of silk from different instars of *Bombyx mori*. *Biomacromolecules*. 2019, 20(3):1203-1216."

- [2] Inoue S, et al. Silk fibroin of *Bombyx mori* is secreted, assembling a high molecular mass elementary unit consisting of H-chain, L-chain, and P25, with a 6:6:1 molar ratio. *J Biol Chem*. 2000, 275(51):40517-40528.
- [3] Tanaka K, et al. Determination of the site of disulfide linkage between heavy and light chains of silk fibroin produced by *Bombyx mori*. *Biochim Biophys Acta*. 1999, 1432(1):92-103.
- [4] Inoue S, et al. Assembly of the silk fibroin elementary unit in endoplasmic reticulum and a role of L-chain for protection of alpha1,2-mannose residues in N-linked oligosaccharide chains of fibrohexamerin/P25. *Eur J Biochem*. 2004, 271(2):356-366.
- [5] Inoue S, et al. A fibroin secretion-deficient silkworm mutant, Nd-sD, provides an efficient system for producing recombinant proteins. *Insect Biochem Mol Biol*. 2005, 35(1):51-59.
- [6] Cui Y, et al. New insight into the mechanism underlying the silk gland biological process by knocking out fibroin heavy chain in the silkworm. *BMC Genomics*. 2018, 19(1):215.
- [7] Ye X, et al. Mechanism of the growth and development of the posterior silk gland and silk secretion revealed by mutation of the fibroin light chain in silkworm. *Int J Biol Macromol*. 2021, 188:375-384.
- [8] Wu M, et al. P25 gene knockout contributes to human epidermal growth factor production in transgenic silkworms. *Int J Mol Sci*. 2021, 22(5):2709.
- [9] Wang F, et al. Protein composites from silkworm cocoons as versatile biomaterials. *Acta Biomater*. 2021, 121:180-192.
- [10] Kunz RI, et al. Silkworm sericin: properties and biomedical applications. *Biomed Res Int*. 2016, 2016:8175701.

Major comment 3: As the authors have shown that SER3 has cysteine residue, why did SER3 not interact with Fib-H or Fib-L via disulfide bonds instead of an independent SM formation?

Response: Thank you for your comments. As you mentioned, the recombinant SER3 has the possibility of direct binding to FIB-H or FIB-L via disulfide bonds. However, as shown in Fig. 2b and Fig. 4a, fusion protein SER3 expressed in the PSG of mutant silkworm is unevenly distributed in the fibroin and formed independent liquid minisomes (recombinant SER3 protein minisomes, SM), indicating that they polymerize more easily. This result is supported by the laser confocal micrograph of silk fiber (revised Fig. 2a) and the transmission electron micrograph of silk fiber section (revised Fig. 2c, Fig.S3a & S3b), and the fluorescence micrograph in the lumen of silk gland also supports the existence of SM (revised Fig. 4a). The possible reason is that the recombinant SER3 has strong hydrophilicity and is not easy to bind to hydrophobic Fib-H or Fib-L [3-5].

In the manuscript, to enhance the expression and secretion of SER3 protein by PSG cells, the *Fib-H* gene promoter sequence and the signal peptide were introduced upstream of the *Ser3* gene sequence. The *EGFP* reporter gene sequence and the 333 bp base sequence at the 3' end of the *Fib-H* gene were connected downstream of the *Ser3* gene sequence. The N-end of the fusion protein contains cysteine has the possibility of combining FIB-L. Meanwhile, the P25 can combined with the NTD of recombinant SER3 protein, like combined with the Fib-H. On the other hand, it has been reported that SER3 in silk gland or cocoon silk has the phenomenon of post-translational modification, which is easy to form dimer [5]. Our western blotting results also confirmed that the recombinant SER3 was similar to the natural SER3 and appeared mainly in the cocoon silk as a dimer (Fig. 2f).

References

- [1] Tanaka K, et al. Determination of the site of disulfide linkage between heavy and light chains of silk fibroin produced by *Bombyx mori*. *Biochim Biophys Acta*. 1999, 1432(1):92-103.
- [2] Inoue S, et al. Assembly of the silk fibroin elementary unit in endoplasmic reticulum and a role of L-chain for protection of alpha1,2-mannose residues in N-linked oligosaccharide chains of fibrohexamerin/P25. *Eur J Biochem*. 2004, 271(2):356-366.
- [3] Takasu Y, et al. Identification and characterization of a novel sericin gene expressed in the anterior middle silk gland of the silkworm *Bombyx mori*. *Insect Biochem Mol Biol*. 2007, 37(11):1234-1240.
- [4] Liu R, et al. Insights into regulatory characteristics of the promoters of Sericin 1 and Sericin 3 in transgenic silkworms. *Biochem Biophys Res Commun*. 2020, 522(2):492-498.
- [5] Hao Z, et al. New insight into the mechanism of *in vivo* fibroin self-assembly and secretion in the silkworm, *Bombyx mori*. *Int J Biol Macromol*. 2021, 169:473-479.

Major comment 4: Why was P25 located between fibroin and sericin?

Response: Thank you for your comments. We found this interesting phenomenon, and the detailed mechanism is worthy of further study in the follow-up study. The related analysis has been added to the discussion in the revised manuscript.

As you know, P25 is a glycoprotein containing Asn-linked oligosaccharide chains and forms a compact structure due to intramolecular disulfide linkages but associates with the H-L complex by non-covalent interactions [1]. The force of this hydrophobic interactions is much weaker than that of covalent bonds between Fib-H and Fib-L. As shown in Fig. 1c, the recombinant protein SER3 has NTD of Fib-H, which can be connected to P25 by non-covalent interactions [2] and detach it from the fibroin elementary units.

On the other hand, the recombinant SER3 expressed in the PSG of mutant is unevenly distributed in the fibroin (Fig. 4a). When the silk protein colloid in lumen of PSG enters MSG, some hydrophilic recombinant SER3 will disperse from the fibroin layer and integrate into sericin layer, due to the gradual dehydration of protein colloid or the repulsion of fibroin molecular folding. In the process of P25 dispersing from the fibroin layer to the outer layer, it is difficult to enter the hydrophilic sericin layer due to highly hydrophobic [3], resulting in most of it staying between the fibroin layer and the sericin layer. In fact, EGFP located the recombinant SER3 in the cocoon silk and found that it was more distributed between the sericin layer and the fibroin layer (revised Fig. 2b).

References

- [1] Tanaka K, et al. Determination of the site of disulfide linkage between heavy and light chains of silk fibroin produced by *Bombyx mori*. *Biochim Biophys Acta*. 1999, 1432(1):92-103.
- [2] Inoue S, et al. Assembly of the silk fibroin elementary unit in endoplasmic reticulum and a role of L-chain for protection of alpha1, 2-mannose residues in N-linked oligosaccharide chains of fibrohexamerin/P25. *Eur J Biochem*. 2004, 271(2):356-366.
- [3] Hao Z, et al. New insight into the mechanism of *in vivo* fibroin self-assembly and secretion in the silkworm, *Bombyx mori*. *Int J Biol Macromol*. 2021, 169:473-479.

Major comment 5: How did P25 affect the stability and structure of fibroin?

Response: Some studies have reported the effect of P25 protein on the stability and structure of silk fibroin fiber [1, 2], so we did not describe in the manuscript. In this manuscript, there is no direct result about the P25 affects the stability and structure of fibroin.

In *Bombyx mori*, the silk fibroin is secreted in a form of a 2.3-MDa protein complex designated as the elementary unit of fibroin [3], which consists of six sets of heavy chain (Fib-H chain; 350kDa fibrous protein)-light chain (Fib-L chain; 26 kDa) disulfide-linked heterodimer and one molecule of a glycoprotein, fibrohexamerin (fhx)/P25 [4]. Fhx/P25 contains three N-linked oligosaccharide chains at Asn69, Asn113, Asn133 [3] and exists either in a 30-kDa (major) or 27-kDa (minor) molecular form [3, 5], which has been suggested to have different compositions of oligosaccharide chains [3]. Fhx/P25 associates with (H-L)₆ mainly by hydrophobic interactions [5].

An analysis of the 4th-instar larval silk and the cocoon silk by LC-MS/MS revealed that P25 protein is the main reason for the enhancement of mechanical properties of IV-E silk [6]. A result of recent research shows that P25 is dispensable for silk formation, it contributes to the stability of fibroin complexes during intracellular transport and affects the morphology of fibroin secretory globules in the PSG lumen [2].

References:

- [1] Wu M, *et al.* P25 Gene Knockout Contributes to Human Epidermal Growth Factor Production in Transgenic Silkworms. *Int J Mol Sci.* 2021, 22(5):2709.
- [2] Zabelina V, *et al.* Mutation in *Bombyx mori* fibrohexamerin (P25) gene causes reorganization of rough endoplasmic reticulum in posterior silk gland cells and alters morphology of fibroin secretory globules in the silk gland lumen. *Insect Biochem Mol Biol.* 2021, 135, 103607.
- [3] Inoue S, *et al.* Silk fibroin of *Bombyx mori* is secreted, assembling a high molecular mass elementary unit consisting of H-chain, L-chain, and P25, with a 6:6:1 molar ratio. *J Biol Chem.* 2000, 275(51):40517-40528.
- [4] Chevillard M, *et al.* Complete nucleotide sequence of the gene encoding the *Bombyx mori* silk protein P25 and predicted amino acid sequence of the protein. *Nucleic Acids Res.* 1986, 14, 6341–6342.
- [5] Tanaka K, *et al.* Hydrophobic interaction of P25, containing Asn-linked oligosaccharide chains, with the H-L complex of silk fibroin produced by *Bombyx mori*. *Insect Biochem. Mol. Biol.* 1999, 29, 269–276.
- [6] Peng Z, *et al.* Structural and mechanical properties of silk from different instars of *Bombyx mori*. *Biomacromolecules.* 2019, 20(3):1203-1216.

Major comment 6: The authors have shown that SM is free or independent from fibroin. SER3 is predicted to have α -helix structure without β -sheet. How did SER3 expression result in an increase in β -sheet content of silk fiber?

Response: Thank you for your comments. According to your major comment 13, we used the same peak number for curve fitting and secondary structure determination of FTIR data in original Fig. 2. The results showed that the β -sheet content was 47.69%

in the WT group and 47.88% in the SER3 group. The difference between the two groups did not appear to be significant.

In the revised version, the FITR analysis data and expression have been modified, and the FITR results of the original Fig. 2e & 2f have been moved to supplementary information.

Major comment 7: Usually, the amorphous region formed by α -helix and random coil structure determines the elasticity and strain of animal silk. It is doubtful that the decrease in α -helix content will not affect the strain (Figure 3c).

Response: The mechanical properties of protein fiber and its α - Helix and β - Sheets are not completely linear. The stress of Spider silk is higher than *Bombyx mori*, but its α -helix content and β - sheets content is lower than silkworm [1].

In this manuscript, the addition of recombinant SER3 changes the protein composition and structure of silk fibroin microfibrils, the crystallinity of silk fibers increased after investigated by X-ray diffraction, which may be the basis for changing its mechanical properties.

References:

- [1] Stengel D, Addison JB, Onofrei D, et al. Hydration-Induced β -Sheet Crosslinking of α -Helical-Rich Spider Prey-Wrapping Silk. *Advanced Functional Materials*. 2021, 31, 2007161.

Major comment 8: Line 256, “Moreover, the sericin microsomes dispersed in the fibrils significantly improved the moisture absorption and liberation of the silk fibers, thereby improving the performance of the textile material.” In fact, the sericin microsomes were dispersed in the fibrils, which may disrupt the structure of mutant silk. How does it improve the mechanical properties of mutant silk?

Response: Thank you for your comments. We have revised the related description in the revision.

As you pointed out, the sericin microsomes changed the original structure of wild type silk, resulting in the distribution of part of P25 protein on the periphery of fibroin (Fig. 2c). The current research shows that the deletion of Fib-H and Fib-L will lead to the inability of silk gland to secrete fibroin [1-3], while the deletion of P25 will not affect the formation of fibroin fiber [4-5].

The colloidal sericin microsomes are distributed between the fibrotic fibrils of the cocoon silk, which will change the arrangement gap of the fibrils and the sliding performance of the fibrils, but will not disrupt the fibrils.

The change of moisture absorption and moisture liberation of cocoon silk is related to the sericin existing between the fibrils.

(1) Sericin is rich in serine and aspartic acid, and has strong moisture absorption and liberation properties due to more hydrophilic groups in the side chain [6-7]. As shown in Fig. 2b and Fig.4a, fibroin contains a certain amount of sericin (recombinant SER3), which helps to improve the moisture absorption and liberation of the silk fibers.

(2) As shown in revised Fig. 2b, the recombinant SER3 in the cocoon silk was more distributed between the sericin layer and the fibroin layer. Hence, the sericin microsomes on the superficial surface of silk fibrils may fall off by degum of cocoon silk and leave tiny gaps between fibrils, thus affecting the moisture absorption and desorption properties of silk.

The Original: Moreover, the sericin microsomes dispersed in the fibrils significantly improved the moisture absorption and liberation of the silk fibers, thereby improving the performance of the textile material.

Revise: Moreover, in view of the improved moisture absorption and liberation of the silk fibers, thereby improving the performance of the textile material.

References

- [1] Inoue S, et al. A fibroin secretion-deficient silkworm mutant, Nd-sD, provides an efficient system for producing recombinant proteins. *Insect Biochem Mol Biol.* 2005, 35(1):51-59.
- [2] Cui Y, et al. New insight into the mechanism underlying the silk gland biological process by knocking out fibroin heavy chain in the silkworm. *BMC Genomics.* 2018, 19(1):215.
- [3] Ye X, et al. Mechanism of the growth and development of the posterior silk gland and silk secretion revealed by mutation of the fibroin light chain in silkworm. *Int J Biol Macromol.* 2021, 188:375-384.
- [4] Wu M, et al. P25 gene knockout contributes to human epidermal growth factor production in transgenic silkworms. *Int J Mol Sci.* 2021, 22(5):2709.
- [5] Zabelina V, et al. Mutation in *Bombyx mori* fibrohexamerin (P25) gene causes reorganization of rough endoplasmic reticulum in posterior silk gland cells and alters morphology of fibroin secretory globules in the silk gland lumen. *Insect Biochem Mol Biol.* 135, 103607 (2021).
- [6] Kunz RI, et al. Silkworm sericin: properties and biomedical applications. *Biomed Res Int.* 2016, 2016:8175701.
- [7] Ji P. Moisture Absorption and Desorption of Polyester and Polyamide Fibres Modified by Sericin. *Journal of Zhejiang Institute of Silk Textiles*, 1994, 11(2) : 21-26.

Major comment 9: In Figure 1b/2b, more evidence such as MS, WB should be provided to claim SM observed in the fibrils.

Response: Thank you for your comments. In the revised manuscript, the confocal results showed that the recombinant SER3 protein was unevenly distributed in the fibrils of the cocoon silk in the form of microsomes (Revised Fig. 2b). WB results showed that the content of SER3 relative to P25 protein in the mutant silk fiber was 4.3 times higher than that in WT group (Revised Fig. 2f). These results further support the existence of microsomes in silk fibers observed in revised Fig. 2c and Fig.S3a-S3b.

Major comment 10: In Figure 2a, an image of silk fiber under white light should be provided to clearly indicate the details of silk fiber.

Response: Thank you for your comments. We have added an image of silk fiber under white light in the revision.

Major comment 11: In Figure 2c, it is undoubtful that the sericin content in silk layer increased as SER3 was overexpressed in the PSG and then secreted into silk fiber. The authors should measure the expression of SER3 in the PSG and the content of SER3, Fib-H, Fib-L and P25 in the silk fiber to better support the claim of high silk yield. However, we also noticed that there is no significant difference in cocoon weight (Figure S2i) and an increase in the cocoon layer ratio (Figure S2j), which indicated a decrease in the pupal weight of SER3 silkworm. Hence, the high silk yield via ectopic expression of SER3 in the PSG is not convincing.

Response: Thank you for your comments. The expression of recombinant *Ser3* gene was measured and reported in Fig. 4e. In the revision, the western blotting was used to determine the SER3 content in cocoon silk fiber with P25 as internal reference (Fig. 2f).

As you pointed out that there is no significant difference in cocoon weight between the mutant (1.002 ± 0.067) and WT (0.982 ± 0.070) (Fig.S2i). However, the PSG/SG parameter representing the development of the posterior silk gland in the SER group (Fig. S2c) was higher than that in the WT group, thus suggesting a potential advantage in the accumulation of silk material in the posterior silk gland of SER during the larval stage. The cocoon layer weight of SER was 116.8% of that of WT group, from 0.104 g per cocoon in WT group to 0.123 g in SER group (revised Fig. S2k). The cocoon layer rate (cocoon silk production efficiency) of the SER silkworm was 114.8% of that of the WT, from 10.64% in WT group to 12.22% in SER group (revised Fig. S2l). Meanwhile, the pupal weight of SER group was 0.880g (0.880 ± 0.062) per pupa in WT group decreased to 0.877g (0.877 ± 0.061) in SER group, no significant difference (revised Fig. S2j).

Fig. S2 revision. The mutant SER silkworm growth and cocoon silk production efficiency. (c) Complete silk glands were dissected and weighed to calculate the ratio of posterior silk gland weight to silk gland weight (PSG/SG). At 72 h after cocooning, 31 cocoons of the same sex (female) were randomly selected and weighed to calculate the percentage of cocoon shell weight in the cocoon weight (cocoon layer rate). (i) Cocoon weight, (j) Pupal weight, (k) Cocoon layer weight, and (l) Cocoon layer rate.

Major comment 12: In Figure 2c, the authors stated that the percentage of sericin in cocoon silk in the SER group was 7.39% higher than that in the WT group, an increase

in 21.8%. How did the authors calculate the percentage of sericin in cocoon in SER3 group and make the comparison with WT group? I suggest they perform SDS-PAGE and western blot to clearly show the expression of SER3 in the fibroin layer, but not the actual yield of silk protein.

Response: Thank you for your comments. We have revised in the revision.

Using a classical degumming method of cocoon silk to determine the sericin content, the percentage of sericin in the SER group was 40.77%, and was 7.39% higher than that of 33.38% in the WT group (Revised Fig. 2e).

In the revision, the western blotting was used to determine the SER3 content in cocoon silk with P25 as internal reference. The results showed that the SER3 content in mutant was 4.3 times higher than that in WT group (Revised Fig. 2f).

Major comment 13: In Figure 2e-f, the result is not convincing as different peak numbers were applied for the curve-fitting and the determination of the secondary structure. Also, the same wavenumber was assigned to different secondary structure. In fact, according to the data provided by the authors, the curve-fitting results using a general procedure showed that the β -sheet content was 42.05% in the WT group and 44.38% in the SER3 group. The difference between the two groups did not appear to be significant.

Response: Thank you for your correction. According to your suggestion, we used a same peak number for curve fitting and secondary structure determination of FITR data. The results showed that the β -sheet content was 47.69% in the WT group and 47.88% in the SER3 group. The difference between the two groups did not appear to be significant.

In order to analyze the microstructure of mutant silk more comprehensively, we performed SAXS and WAXD for characterizing silk crystal structure and size. The results showed that the crystallinity of fibroin fibers in SER group was increased and the mutant and WT cocoon silk have different electron density in the crystalline and amorphous regions of the periodic structure (e.g. fibroin fibrils) at the nano scale.

In the revised version, the FITR analysis data has been replaced by SAXS and WAXS results, and the FITR results of the original Fig. 2e & 2f have been moved to supplementary information.

Major comment 14: The authors should provide more evidence that the silk structure has been indeed changed. I suggest to perform SAXS or WAXS for characterizing silk crystal structure and size.

Response: Thank you for your constructive suggestion. We have performed SAXS and WAXD and the related results and analysis have been added to the appropriate position in the revised manuscript.

In the revision, a D8 Advance X-ray diffractometer was used to identify the crystalline phase in the fibroin fibrils samples. According to the crystal peak position of cocoon silk, the crystal diffraction peaks of silk fibroin fibers were detected at approximately 9.0°, 20.4° and 29.1° (Fig.3k). The calculated relative crystallinity results showed that the crystallinity of fibroin fibers in WT and SER groups was 36.62%

and 42.29% respectively (Fig.3l), thus indicating greater crystallinity of fibroin fibers in the SER group.

SAXS test results revealed two-dimensional images close to a double wedge shape (Fig.3m), in which the short diameter in the SER group was longer than that in the WT group, thus indicating that both SER and WT cocoon silk fibers are anisotropic, but the electron density changes before and after X-ray transmission of the two materials differed. The scattering intensity curve showed a significant difference in discrete intensity in the angle range of angle 0.1° - 0.6° (Fig.3n), thus indicating that the mutant and WT cocoon silk differed in electron density in the crystalline and amorphous regions of the periodic structure (e.g., fibroin fibrils) at the nanoscale.

Major comment 15: As shown in Figure 4c, it is strange that Fib-H/L/P25 appeared in MP. I suggest the authors should take care the part of silk gland for PCR. Also, there is a significant difference between the relative expression of EGFP and SER3. Could the authors explain the difference in the expression of EGFP and SER3 since they are fused together?

Response: The silk gland (SG) of silkworm is divided into anterior silk gland (ASG), middle silk gland (MSG) and posterior silk gland (PSG) according to different morphology and function. Previous studies have suggested that the proteins of fibroin and sericin, which are produced in the PSG and MSG, respectively [1]. The sericin gene *Ser3* is naturally expressed in the anterior section of MSG of 5th instar larvae [2,3], while *Fib-H* is mainly expressed in the PSG [1, 3]. However, so far, there is not strictly identified the expression regions of a variety of silk fibroin and sericin in SG.

In this study, according to the schematic diagram of Fig. 4a, MSG was divided into three regions, the anterior section of MSG (MA), the middle section of MSG (MM) and the posterior section of MSG (MP) respectively; PSG was divided into two regions, the anterior section of PSG (PA) and the posterior section of PSG (PP). When PCR testing, we strictly sampled the intermediate area of MA, MM, MP, PA, or PP to avoid interference from adjacent silk gland tissue. Our results showed that the fibroin genes *Fib-H*, *Fib-L*, and *P25* were not only expresses high in the entire PSG (PA and PP) of WT silkworm, but also has a slight expression in MP (Fig. 4c). The Fib-H/L/P25 appeared in MP can be verified in multiple silkworm varieties, which is a reliable result. Hence, *EGFP* and *Ser3* genes driven by *Fib-H* promoter were not only expresses high in PA and PP of SER silkworm, but also has a slight expression in MP (Fig 4c, 4d & 4e).

Using different primers to PCR a gene from the same sample, the electrophoretic bands of PCR products will be different due to the difference of optimal PCR conditions. The *EGFP* and *Ser3* in MP are transcripts of a fusion genes, there is no difference (or minimal difference) in the copy number of mRNA in theory, but their PCR primers are different, the optimal PCR conditions are different, and it is normal for the electrophoresis bands of PCR products to differ. According to the results of Fig. 4c-4e, the recombinant gene *EGFP* and *Ser3* driven by *Fib-H* promoter were not be expressed in MA and MP, but were expressed in PP, PA and MP. Meanwhile the expression level of the *EGFP* and *Ser3* both decreased successively in PP-PA-MP, and the expression trend was the same as *Fib-H* gene.

References

- [1] Xia Q, et al. Advances in silkworm studies accelerated by the genome sequencing of *Bombyx mori*. *Annu Rev Entomol*. 2014, 59:513-536.

- [2] Takasu Y, et al. Identification and characterization of a novel sericin gene expressed in the anterior middle silk gland of the silkworm *Bombyx mori*. *Insect Biochem Mol Biol*. 2007, 37(11):1234-1240.
- [3] Dong Z, Zhao P, Wang C, Zhang Y, Chen J, Wang X, Lin Y, Xia Q. Comparative proteomics reveal diverse functions and dynamic changes of *Bombyx mori* silk proteins spun from different development stages. *J Proteome Res*. 2013, 12(11):5213-22.

Major comment 16: According to the results provided by the authors, the silk structure of the mutant was changed dramatically. Then the author should analyze the microstructure of mutant silk more comprehensively, as FTIR is usually considered as a semi-quantitative method. Wide-angle X-ray diffraction or small-angle X-ray scattering could clearly characterize the crystallinity, grain size, orientation and other key structural information of polymer materials, which should be supplemented to better indicate the structural changes of mutant silk.

Response: Thank you for your comments. The SAXS and WAXD was performed for characterizing silk crystal structure and size.

In the revision, a D8 Advance X-ray diffractometer was used to identify the crystalline phase in the fibroin fibrils samples. According to the crystal peak position of cocoon silk, the crystal diffraction peaks of silk fibroin fibers were detected at approximately 9.0°, 20.4° and 29.1° (Fig.3k). The calculated relative crystallinity results showed that the crystallinity of fibroin fibers in WT and SER groups was 36.62% and 42.29% respectively (Fig.3l), thus indicating greater crystallinity of fibroin fibers in the SER group.

SAXS test results revealed two-dimensional images close to a double wedge shape (Fig.3m), in which the short diameter in the SER group was longer than that in the WT group, thus indicating that both SER and WT cocoon silk fibers are anisotropic, but the electron density changes before and after X-ray transmission of the two materials differed. The scattering intensity curve showed a significant difference in discrete intensity in the angle range of angle 0.1° -0.6° (Fig.3n), thus indicating that the mutant and WT cocoon silk differed in electron density in the crystalline and amorphous regions of the periodic structure (e.g., fibroin fibrils) at the nanoscale.

Major comment 17: It is suggested that proteomic analysis of degumming mutant silk should be performed to precisely determine SER3 content in silk fiber, which is essential to address the mechanism by which SER3 affects the structure and properties of silk.

Response: Thank you for your suggestion. In the revision, we do not provide proteomic data, the western blotting was used to determine the SER3 content in cocoon silk with P25 as internal reference. The results showed that the SER3 content in mutant was 4.3 times higher than that in WT group (Revised Fig. 2f). As shown in Fig.S3, the laser confocal microscope clearly observed that the EGFP labeled recombinant SER3 protein was still widely distributed in the degummed silk fibers. The results in Fig. 2b revision showed more clearly that the recombinant SER3 protein appeared in the fibroin area as particles of different sizes. The measurable particle size was 0.05-0.50 μm,

which was unevenly distributed, and the most distribution was between the fibroin layer and sericin layer, secondly is among the fibrils of silk fibroin.

Major comment 18: The author mentioned that the number of cocoons for tensile testing was 20. In the data provided by the authors, the number of samples was 22 (SER) and 27 (WT). It can be assumed that only 1 or 2 silks strands from each cocoon are used for testing. Due to the obvious variance in the mechanical properties of silk, this method does not seem to accurately reflect the overall silk properties.

Response: We are very sorry that is not clearly described in the method, which makes you confused. The method has been modified in the revised manuscript.

As you indicate, the mechanical properties of silk fibers in different parts of a cocoon are quite different [1]. In order to reduce this effect, the mechanical properties of raw silk are determined according to the national standard of China [GB/T 1798-2008 Testing method for raw silk]. Cocoons (20 cocoons of WT or SER) were boiled in water and fully expanded before reeling to obtain raw silk. The reeling cocoon number per raw silk is 10 cocoons, and the reeling wire speed is 44-46 m/min. The obtained raw silk fiber retained most of sericin, and one sample was taken every 3 meters approximately between 100-200 meter to determine the mechanical properties or diameter of the silk. 22 samples were measured to determine the mechanical properties in SER group and 27 samples were measured in WT group.

References

- [1] Qu J, et al. Study on the effect of stretching on the strength of natural silk based on different feeding methods. *ACS Biomater Sci Eng.* 2022, 8(1):100-108.

Major comment 19: Generally, a length of 100 mm and a tensile speed of 100 mm/min are applied for mechanical performance test. It is obvious the length and tensile speed may affect the mechanical performance of silk fiber. Could the authors please explain why did you perform the test with the parameters different from literatures?

Response: As you pointed out that the length and tensile speed may affect the mechanical performance of silk fiber. Different parameters are used in different studies [1,2,3].

In this study, an initial length of 250 mm and a tensile speed of 250 mm/min are applied for mechanical performance test, which is referred the method of Wang et al. (2009). Since the mechanical properties of silk fibers in this paper are only used for the comparison of relative values between SER and WT groups, the parameters obtained by using the same fiber length and tensile speed will not affect the comparison results.

References

- [1] Wang S, et al. Structure and properties of naturally colored silk. *Journal of Textile Research.* 2009. ,30(11):5-9. DOI:10.13475/j.fzxb.2009.11.007. (in Chinese)
- [2] Peng Z, et al. Structural and mechanical properties of silk from different instars of *Bombyx mori*. *Biomacromolecules.* 2019, 20(3):1203-1216

[3] Zhang L, et al. The variability of mechanical properties and molecular conformation among different spider dragline fibers. *Fibers and Polymers*. 2013,14(7):1190-119.

Major comment 20: The diameter and cross-sectional area of the silk may affect the stress. Hence, the authors should provide the diameter and the cross-sectional area of silk and indicate how the cross-sectional area is determined.

Response: As you pointed out, the diameter and cross-sectional area of the cocoon silk may affect the stress. However, the stress is obviously affected by the fiber structure and protein composition of cocoon wires.

Results in Fig. S3g & S3h are diameter and cross-sectional area of the silk. The diameter of the silk sample was measured using a digital microscope at 1000× magnification; multiple measurements were obtained from each sample, and the average diameter was calculated, then obtained the cross-sectional area ($S = \pi(d/2)^2$).

4) writing issues

1. The abstract should be well revised to better indicate the most important findings and the significance of this study. For example, the outer layer sericin SER3 was ectopically expressed in the PSG of the silkworm via a piggyBac-mediated transgenic approach, then secreted into the inner fibroin layer, thus generating a new fiber with sericin microsomes dispersed in fibroin fibrils. The cause and effect in the sentence “Moreover, the water solubility and stability of the fibroin-colloid in the silk glandular cavity are increased, thus significantly improving the β -sheet content of fibroin, as well as the mechanical properties, moisture absorption and moisture liberation of the silk fiber” is not valid.

Response: Thanks for your correction. We have made some changes to the abstract in the revision.

2. The introduction should be well revised to clearly indicated the purpose, contents and significance of this study. I'm confused about the mechanism of the metastability of ultra-high concentration aqueous solutions of Fib-H/Fib-L/P25 polymers in SGs, or altering the ancient silk structure via innovative reprogramming of the genomes of SG cells with high survival rate and silk yield. It is difficult to understand the relationship between the sentences “the fibril structure and function of the ancient silk fiber were greatly altered” and “This method may help address the bottleneck problems of the low survival rate and low silk yield of genetically transgenic silkworms”. Also, I'm confused that the function of the ancient silk fiber was greatly altered. What is the function of the ancient silk fiber and how the function of the fiber was changed?

Response: Thank you for your comments. We have made some changes to the introduction in the revision.

The modern silk still retains the molecular structure of ancient silk, although it has been thousands of years since silkworm was used to produce silk. We use the word

‘ancient silk’ by borrowing the usage of Omenetto and Kaplan (Science, 2010) [1] on silkworm cocoon silk.

At present, more than a thousand silkworm varieties have been selected and bred, but modern silk still retain the molecular structure of wild silkworm cocoon silk. The silk fiber of silkworm cocoon has a core-shell type structure, with silk fibroin as the inner core and sericin as the outer coating. Each silk fibroin brin is composed of numerous interlocking fibroin fibrils [2]. Fibroin is the main component, accounts for > 70% of cocoon silk proteins, and is composed of Fib-H, Fib-L, and P25 proteins in a 6:6:1 molar ratio [1-5].

References

- [1] Omenetto FG, Kaplan DL. New opportunities for an ancient material. *Science*. 2010, 329(5991):528-531.
- [2] Huang W, et al. Silkworm silk-based materials and devices generated using biotechnology. *Chem Soc Rev*. 2018, 47(17):6486-6504.
- [3] Li G, et al. Silk-based biomaterials in biomedical textiles and fiber-based implants. *Adv Healthc Mater*. 2015, 4(8):1134-1151.
- [4] Hao Z, et al. New insight into the mechanism of *in vivo* fibroin self-assembly and secretion in the silkworm, *Bombyx mori*. *Int J Biol Macromol*. 2021, 169:473-479.
- [5] Inoue S, et al. Silk fibroin of *Bombyx mori* is secreted, assembling a high molecular mass elementary unit consisting of H-chain, L-chain, and P25, with a 6:6:1 molar ratio. *J Biol Chem*. 2000, 275(51):40517-40528.
- [6] Xiang H, et al. The evolutionary road from wild moth to domestic silkworm. *Nat Ecol Evol*. 2018, 2(8):1268-1279.

3. In Figure 1a, spinning dope may be redundantly labeled.

Response: Accepted. We have revised in the revision.

4. In Figure 1c, PiggBac should be piggyBac.

Response: Accepted. We have revised in the revision.

5. The gene names in the Figure 1c, Figures legends (line 489–496), and Materials and Methods (line 300) are not italics; the mutant strain name, gene name and protein name (SER, SER I, SER II, SER III, SER3, ser and ser3) are confused in the manuscript.

Response: Thanks for your correction. In the revision, the mutant was still used as SER.

The natural sericin 3 protein and gene were expressed as SER3 and *Ser3* respectively, and the recombinant *Ser3* gene and its protein were expressed as recombinant *Ser3* gene and the recombinant SER3 protein respectively.

6. In Figure 2a, the abbreviations of fibroin layer, SF and F should be unified.

Response: Thanks for your correction. We have revised in the revision.

7. In Figure 3, The sample was a cocoon silk fiber (100–200 meters) without the sericin protein of the outer layer removed (n=20 cocoons). I am confused about the presence or absence of sericin protein.

Response: As response in ‘Major comment 18’, the mechanical properties of silk fibers in different parts of a cocoon are quite different. In order to reduce this effect, the

mechanical properties of raw silk are determined according to the national standard of China [GB/T 1798-2008 Testing method for raw silk]. Cocoons (20 cocoons of WT or SER) were boiled in water and fully expanded before reeling to obtain raw silk with a target size of 20/22 dtex. The obtained raw silk fiber retained most of sericin, and one sample was taken every 3-4 m between 100-200 m to determine the mechanical properties. 22 samples were measured in the SER group and 27 samples were measured in the WT group.

8. Figure 3 k-n are not closely related with the mechanical performance of silk fiber and should be removed or stated in the supplementary materials.

Response: Accepted. We have moved the results to the supplementary materials (Fig. S4) and stated it in the discussion.

9. Figure 4a should be well re-organized to clearly indicate the different part of silk gland.

Response: Accepted. We have revised in the revision.

10. Figure S2, figure and figure legend of c, d, e and f are not appropriate.

Response: Thanks for the corrections. We have revised in the revision.

11. Figure S3c, dvunitka → ultra-dense?

Response: Thanks for the corrections. We have revised in the revision.

12. Line 366, μl → μL

Response: Accepted. We have revised in the revision.

13. Line 65-67, the author mentioned the reprogramming of the genomes of SG cells, which is irrelevant to the topic of this study and should therefore be deleted.

Response: Accepted. We have revised in the revision.

14. The manuscript should be well proof edited by a native English speaker to polish the grammar, expression and organization and correct the typos.

Response: Accepted. Thank you again for correcting the language and grammar of the manuscript. The revision was edited by a native English speaker from International Science Editing Scientific Services.

Reviewer #3:

This manuscript describes the production of a novel type of *B. mori* silk fiber with different molecular compositions and fiber morphology from normal silk using a transgenic technology. The authors showed that such differences have led to better mechanical and moisture absorption properties of silk. The effects of transgenesis on silk's properties are unique and might be useful in practical applications. Therefore, this manuscript would have considerable impacts on the researchers in the field of proteins materials. However, some of the authors' conclusions are not fully supported by the data in the manuscript. This manuscript is thus not appropriate for publication in Nature Communications in the present form.

Response: Thank you very much for your comments concerning our manuscript. Those comments are all valuable and very helpful for revising and improving our manuscript, as well as the important guiding significance to our researches. We have tried our best to improve the manuscript and have made a lot of changes which we hope meet with approval.

Major issues:

Major comment 1: Line 72-79. The authors describe many negative aspects of previous genetic alterations of silkworms. Although there are many successful examples of genetic alterations, they seem to emphasize negative aspects of genetic engineering too much by introducing some specific examples such as expressing "cytotoxin" in PSG (Ref 29). The authors should summarize previous studies in a fair position.

Response: Thanks for your comments. We have revised in the revision. As shown in **Table S1**, *Bombyx mori* expressed exogenous protein with molecular weight greater than 100 kDa in its silk gland, which was prone to silk gland development deformity and decreased individual survival rate, and the cocoon silk production efficiency was significantly reduced, resulting in thin layered cocoon shells [1-4]. In the existing reports that though the cocoon silk yield is not abnormal, the expression of foreign proteins is generally not high. The highest content of foreign proteins reported is only 1.1% of the cocoon silk weight, of which the expression in the posterior silk gland is less than 0.84% of the cocoon silk [5-7]. It shows that the silk gland of silkworm is a highly specialized self-silk protein expression tissue, and the function of expressing foreign proteins needs to be improved.

References

- [1] Otsuki. R. et al., Bioengineered silkworms with butterfly cytotoxin-modified silk glands produce sericin cocoons with a utility for a new biomaterial. **Proc. Natl. Acad. Sci. USA** 114, 6740-6745 (2017).
- [2] Minagawa, S. et al. Production of a correctly assembled fibrinogen using transgenic silkworms. **Transgenic Res** 29, 339-353 (2020).
- [3] Wang H, et al. High yield exogenous protein HPL production in the *Bombyx mori* silk gland provides novel insight into recombinant expression systems. **Sci Rep.** 5:13839 (2015).
- [4] Teulé, F. et al. Silkworms transformed with chimeric silkworm/spider silk genes spin composite silk fibers with improved mechanical properties. **Proc. Natl. Acad. Sci. USA** 109, 923-928 (2012)
- [5] Kuwana, Y. et al. High-toughness silk produced by a transgenic silkworm expressing spider (*Araneus ventricosus*) dragline silk protein. **PLoS One** 9,

e105325 (2014)

- [6] Iizuka, M. et al. Production of a recombinant mouse monoclonal antibody in transgenic silkworm cocoons. **FEBS. J** 276, 5806-5820 (2009)
- [7] Tomita, M. et al. Transgenic silkworms produce recombinant human type III procollagen in cocoons. **Nat. Biotechnol** 21, 52-56 (2003)

The Original: However, the efforts to express and secrete exogenous proteins in the SGs of silkworms through transgenic technology to date have generally resulted in low efficiency silk protein synthesis and secretion, at expression levels far below those of normal silk proteins (Supplementary, Table 1). The performance of new structural silk produced by transgenic silkworms is far inferior to that of fibers produced by donor animals. Moreover, the problems of SG deformity and declining silk production efficiency are common^{29,30}. The strategy of directly introducing functional silk protein genes from other organisms or similar artificially designed genes into the silkworm genome to produce new silkworm silk in the SG has been problematic.

Revise: Although the efforts to express and secrete exogenous proteins in the SGs of silkworms through transgenic technology to date have many successful examples of genetic alterations. However, it is still a great challenge to greatly improve the expression efficiency of foreign proteins while maintaining the cocoon silk yield, especially to express high molecular weight proteins (~100 kDa) in the posterior silk glands^{23,24,28-32}.

Major comment 2: Line 82-83. This sentence would cause misunderstanding that the present genetic engineering method is not suitable for practical applications. There are actually some trials for commercial productions of genetically engineered silks in some countries. It is also unclear how the method proposed in this manuscript can solve the bottlenecks shown here. Therefore, this sentence should be omitted or revised. Moreover, the meaning of “low survival rate” is unclear, and “low silk yield” is not observed in many cases except for some specific examples such as ref 29.

Response: Accepted. This sentence ‘This method may help address the bottleneck problems of the low survival rate and low silk yield of genetically transgenic silkworms’ has been deleted in the revised version. The description of low survival rates has been revised in the revised manuscript and replied in **major comment 1**.

Major comment 3: Figure 1b. Sericin II (may be equal to Ser2) is reported to be major coating proteins of larval silk threads spun during the growing stages (Takasu et al. *Insect Biochem Mol Biol.* 2010 Apr;40(4):339-44. doi: 10.1016/j.ibmb.2010.02.010). In addition, I am not sure that the layered structure of sericin has been experimentally verified so far. Please add some appropriate references for this model.

Response: The **Fig. 1a** is drawn according to the layered structure of sericin in **Figure R3**. The **Fig. 1a** and the related description has been revised in the revision.

Although there is no recent research report, the sericin of cocoon silk has been divided into two or three types according to the water solubility and isoelectric point of sericin in early studies. Depending on its solubility sericin is categorized into three fractions: sericin A, sericin B, and sericin C. The outermost layer is sericin A, the middle layer is called sericin B, and the innermost layer is called Sericin C [1-2].

The lumen of the MSG contains more than seven major sericin, as well as various uncharacterized minor proteins, which are encoded mainly by three genes: *Ser1*, *Ser2*, and *Ser3* [3]. During the 5th instar larval stage, *Ser1* is highly expressed in the middle part of the middle silk gland (M-MSG) and the posterior part of the middle silk gland (P-MSG), *Ser2* is only expressed in the anterior part of the middle silk gland (A-MSG), while *Ser3* are expressed in A-MSG and M-MSG [4]. The water solubility of sericin in the outer layer is significantly higher than that in the inner layer. The outer layer is mainly composed of sericin 3 and the inner layer is mainly composed of sericin 1, even though the sericin composition in the middle layer is uncertain [5].

However, as you pointed out, *Ser2* is reported to be major coating proteins of larval silk threads spun during the growing stages, rather than the main sericin component in cocoon silk [6-8]. Quantitative proteomic analysis showed that sericin in cocoon silk was mainly sericin 1, sericin 3 content was about 2.5% of sericin1, and sericin 2 content was only 5 / 100000 of sericin 1 [6,9].

Figure R3. A representative schematic of silk protein synthesis and secretion in the silk glands of *Bombyx mori* [3].

Editorial note: Used with permission of Annual Reviews, Inc, from Annual review of entomology, Xia, Q., Li, S., & Feng Q, 59, 2014; permission conveyed through Copyright Clearance Center, Inc.

References

- [1] Shaw JT, Smith SG. Amino-acids of silk sericin. *Nature*, 1951, 168 (4278):745
- [2] Sprague KU. The *Bombyx mori* silk proteins: characterization of large polypeptides. *Biochemistry*. 1975, 14(5):925-931.
- [3] Xia Q, et al. Advances in silkworm studies accelerated by the genome sequencing of *Bombyx mori*. *Annu Rev Entomol*. 2014, 59:513-536.
- [4] Li H, et al. Identification and location of sericin in silkworm with anti-sericin antibodies. *Int J Biol Macromol*. 2021, 184:522-529.

- [5] Huang Guo Rui. Cocoon silk study (National College Textbooks, China) [M]. Beijing: **Agricultural Press**,1994// SILK REELING. Translator D. Mahadevappa. New Hampshire: **Science Publishers, Inc., USA**, 1998
- [6] Takasu Y, et al. Identification of Ser2 proteins as major sericin components in the non-cocoon silk of *Bombyx mori*. **Insect Biochem Mol Biol**. 2010, 40(4):339-344.
- [7] Dong Z, Zhao P, Wang C, Zhang Y, Chen J, Wang X, Lin Y, Xia Q. Comparative proteomics reveal diverse functions and dynamic changes of *Bombyx mori* silk proteins spun from different development stages. **J Proteome Res**. 2013, 12(11):5213-22.
- [8] Peng Z, et al. Structural and Mechanical Properties of Silk from Different Instars of *Bombyx mori*. **Biomacromolecules**. 2019, 20(3):1203-1216.

Major comment 4: Line 99-101. The presence of sericin 3 protein in the microsomes should be verified by immunostaining or fluorescence observation of fused GFP.

Response: Accepted. The confocal microscope observation results of cocoon silk have been listed in the revised Fig. 2, and we have made some changes in the revision.

Major comment 5: Line 101-103. The authors describe that “the production efficiency of cocoon silk was significantly higher than that of the wild type (WT)”. But no data is shown here. Figure S2i and j respectively show total cocoon weights and the ratio of silk in the total cocoon weights including pupa. The authors should clearly show the comparison of silk and silk fibroin production between SER and WT.

Response: Accepted. We have revised in the revision.

As you pointed out that there is no significant difference in cocoon weight between the mutant (1.002 ± 0.067) and WT (0.982 ± 0.070) (Fig.S2i). However, the PSG/SG parameter representing the development of the posterior silk gland in the SER group (Fig. S2c) was higher than that in the WT group, thus suggesting a potential advantage in the accumulation of silk material in the posterior silk gland of SER during the larval stage. The cocoon layer weight of SER was 116.8% of that of WT group, from 0.104 g per cocoon in WT group to 0.123 g in SER group (revised Fig. S2k). The cocoon layer rate (cocoon silk production efficiency) of the SER silkworm was 114.8% of that of the WT, from 10.64% in WT group to 12.22% in SER group (revised Fig. S2l). Meanwhile, the pupal weight of SER group was 0.880g (0.880 ± 0.062) per pupa in WT group decreased to 0.877g (0.877 ± 0.061) in SER group, no significant difference (revised Fig. S2j).

Fig. S2. The mutant SER silkworm growth and cocoon silk production efficiency. (c) Development of the posterior silk gland. After the 5th instar larvae were given mulberry for the first time (0 h), ten larvae of the same sex (male) were randomly selected every 24 h. Complete silk glands were dissected and weighed to calculate the ratio of posterior silk gland weight to silk gland weight (PSG/SG). (i) Cocoon weight, (j) Pupal weight, (k) Cocoon layer weight, and (l) Cocoon layer rate. At 72 h after cocooning, 31 cocoons of the same sex (female) were randomly selected and weighed to calculate the percentage of cocoon shell weight in the cocoon weight (cocoon layer rate).

Major comment 6: Line 105-106. The description “the transgenic silkworm SGs have superior production performance” is not supported by experimental data because no data on silk production is provided.

Response: As shown in revised Fig. S2, there is no significant difference in cocoon weight between the mutant (1.002 ± 0.067) and WT (0.982 ± 0.070) (Fig.S2i). However, the PSG/SG parameter representing the development of the posterior silk gland in the SER group (Fig. S2c) was higher than that in the WT group, thus suggesting a potential advantage in the accumulation of silk material in the posterior silk gland of SER during the larval stage. The cocoon layer weight of SER was 116.8% of that of WT group, from 0.104 g per cocoon in WT group to 0.123 g in SER group (revised Fig. S2k). The cocoon layer rate (cocoon silk production efficiency) of the SER silkworm was 114.8% of that of the WT, from 10.64% in WT group to 12.22% in SER group (revised Fig. S2l).

Major comment 7: Line 118-119. The authors show the percentage of sericin in silk. However, without the amounts of silk and silk fibroin, it is impossible to know whether the increase of the percentage resulted from the increase of sericin production or the decrease of fibroin production.

Response: Using a classical degumming method of cocoon silk to determine the sericin content in cocoon silk, the percentage of sericin in the SER group was 40.77%, and was 7.39% higher than that in the WT group (Revised Fig. 2e).

In the revision, the western blotting was used to determine the SER3 content in cocoon silk with P25 as internal reference. The results showed that the SER3 content in mutant was 4.3 times higher than that in WT group (Revised Fig. 2f).

Major comment 8: Line 119-120. This sentence is not supported without the data of actual amounts of total silk, fibroin, and sericin. SDS-PAGE analysis is also required to show the increase of Ser3 production.

Response: In the revised version, relevant descriptions have been modified.

The Original: The percentage of sericin in cocoon silk in the SER group was 7.39% higher than that in the WT group (Fig. 2c), an increase in 21.8%. Our results indicated that the PSG of the mutant silkworm synthesized the SER3 protein very efficiently and successfully secreted it into the silk fiber.

Revise: Using a classical degumming method of cocoon silk to determine the sericin content in cocoon silk, the percentage of sericin in cocoon silk in the SER group was 7.39% higher than that in the WT group (Fig. 2e), an increase in 21.8%. The western

blotting was used to determine the SER3 content in cocoon silk with P25 as internal reference. The results showed that the SER3 content (native plus recombinant SER3) in mutant was 4.3 times higher than that in WT group (Fig. 2f). Our results indicated that the PSG of the mutant silkworm synthesized the SER3 protein very efficiently and successfully secreted it into the silk fiber.

Major comment 9: Line 161-162. Some quantitative analyses such as molecular weight analysis by SDS-PAGE or GPC are required to conclude that the mutant silk has higher alkali resistance than the WT silk.

Response: We have revised in the revision.

The Original: SEM characterization indicated that the adhesion between silk fibers in the SER cocoon silk layer was closer, and the pores were smaller than those in the WT (Fig. 3i). After removal of sericin with the alkali method, the surfaces of fibroin fibers in the SER group were smoother, and less fibril damage was observed than that in the WT (Fig. 3j). The results showed that the silk fibers in the SER group were more alkali resistant than those in the WT group.

Revise: SEM characterization indicated that the adhesion between silk fibers in the SER cocoon silk layer was closer, and the pores were smaller than those in the WT (Fig. 3i). After removal of sericin with the alkali method, the surfaces of fibroin fibers in the SER group were smoother, and less fibril damage was observed than that in the WT (Fig. 3j). The results showed that degumming had less damage to silk fibers in SER group than in WT group.

Major comment 10: Line 216. Since the authors showed only one example of a mutant silk with changed properties in this manuscript, the word “controllable” is not appropriate here. Please delete it.

Response: Accepted. We have revised in the revision.

Major comment 11: Figure 2e. Since the number of the deconvoluted peaks are different among samples, direct comparison might be inappropriate. More careful discussion is necessary for structural analysis. The assignment of the left-most peaks is different among samples. Please explain why.

Response: Thank you for your comments. We used the same peak number for curve fitting and secondary structure determination of FITR data in original Fig. 2. The results showed that the β -sheet content was 47.69% in the WT group and 47.88% in the SER3 group. The difference between the two groups did not appear to be significant.

In the revised version, the FITR analysis data and expression have been modified, and the Fitr results of the original Fig. 2e & 2f have been moved to supplementary information.

Major comment 12: Line 250. To discuss the change of the structure of silk fibroin, using only an IR analysis is not sufficient. The interpretation of the IR spectra is not

convincing as described in (12). It is preferable to combine with other methods such as X-ray analysis. If the discussion of structural changes is not essential for the conclusions, I recommend that the IR analysis results are omitted.

Response: Thank you for your comments. In the revision, a D8 Advance X-ray diffractometer was used to identify the crystalline phase in the fibroin fibrils samples. According to the crystal peak position of cocoon silk, the crystal diffraction peaks of silk fibroin fibers were detected at approximately 9.0° , 20.4° and 29.1° (Fig.3k). The calculated relative crystallinity results showed that the crystallinity of fibroin fibers in WT and SER groups was 36.62% and 42.29% respectively (Fig.3l), thus indicating greater crystallinity of fibroin fibers in the SER group.

SAXS test results revealed two-dimensional images close to a double wedge shape (Fig.3m), in which the short diameter in the SER group was longer than that in the WT group, thus indicating that both SER and WT cocoon silk fibers are anisotropic, but the electron density changes before and after X-ray transmission of the two materials differed. The scattering intensity curve showed a significant difference in discrete intensity in the angle range of angle 0.1° - 0.6° (Fig.3n), thus indicating that the mutant and WT cocoon silk differed in electron density in the crystalline and amorphous regions of the periodic structure (e.g., fibroin fibrils) at the nanoscale.

Major comment 13: It is preferable to show the data of silk obtained from heterozygous individuals (hybrid of SER and WT lines). If such data is intermediate between SER and WT, the conclusions will be more convincing.

Response: Thank you for your constructive suggestion. As you pointed out, if such data from heterozygous individuals is intermediate between SER and WT, the conclusions will be more convincing. Due to the fact that it takes at least two generations of silkworm rearing to obtain the cocoon silk materials of heterozygotes, the experimental cycle is too long. In addition, due to the impact of the COVID-19 from early March to the end of May this year, silkworm rearing investigation cannot be carried out. In the follow-up study, we will refer to your opinions and further optimize the experiment. In fact, the homozygous SER and WT are very different, and the results in this research is able to support the conclusion that silk fibers are different in groups SER and WT.

Minor issues:

1. The word “piggyBac” should be written in italics.

Response: Thanks for the corrections. We have revised in the revision.

2. Unify the abbreviations of silk fibroin and sericin in Figure 2 (“SF”, “F”, “Fibroin”, “SS”, and “S” are mixed and thus confusing).

Response: Thanks for the corrections. We have revised the Figure 2 in the revision.

3. The words “TALEN” in Line 302 in the main text and Line36 in the SI might be mistyping of “piggyBac”.

Response: Thanks for the corrections. We have revised in the revision.

REVIEWER COMMENTS

Reviewer #1 (Remarks to the Author):

The manuscript has become a better paper after revision. I suggest that the manuscript can be considered to be published at Nature Communications

Reviewer #3 suggested acceptance in comments to the editor